# Structure of the human CTF18–RFC clamp loader bound to PCNA

**Giuseppina R Briola[1†], Mohammad Tehseen[1†], Amani Al-Amodi[1], Grace Young[2], Ammar U Danazumi[1], Phong Quoc Nguyen[1], Christos G Savva[1], Mark Hedglin[2], Samir M Hamdan[1,3]\*, Alfredo De Biasio[1]\***

[1]Bioscience Program, Division of Biomedical Sciences, King Abdullah University of Science and Technology, Thuwal, Saudi Arabia; [2]Department of Chemistry, The Pennsylvania State University, University Park, United States; [3]KAUST Center of Excellence for Smart Health, King Abdullah University of Science and Technology, Thuwal, Saudi Arabia

**\*For correspondence:**
samir.hamdan@kaust.edu.sa (SMH);
alfredo.debiasio@kaust.edu.sa (ADB)

†These authors contributed equally to this work

Competing interest: The authors declare that no competing interests exist.

## eLife Assessment

This paper reports new data on the structure of the human CTF18-RFC clamp loader complex bound to the PCNA clamp. The new and **convincing** data complement previous reports of CTF-RFC-PCNA structures and as such, represents an **important** contribution.

**Abstract** Sliding clamps like PCNA are crucial processivity factors for replicative polymerases, requiring specific clamp loaders for loading onto DNA. The human alternative clamp loader CTF18–RFC interacts with the leading strand polymerase Pol ε and loads PCNA onto primer/template DNA using its RFC pentameric module. Here, we provide a structural characterization of the human CTF18–RFC complex and its interaction with PCNA. Our cryo-EM data support that the Ctf8 and Dcc1 subunits of CTF18–RFC, which form the regulatory module interacting with Pol ε, are flexibly tethered to the RFC module. A 2.9 Å cryo-EM structure shows the RFC module bound to PCNA in an autoinhibited conformation similar to the canonical RFC loader, marking the initial step of the clamp-loading reaction. The unique RFC1 (Ctf18) large subunit of CTF18–RFC, which based on the cryo-EM map shows high relative flexibility, is anchored to PCNA through an atypical low-affinity PIP box in the AAA+ domain and engages the RFC5 subunit using a novel β-hairpin at the disordered N-terminus. We show that deletion of this β-hairpin impairs the CTF18–RFC−PCNA complex stability, slows down clamp loading, and decreases the rate of primer synthesis by Pol ε. Our research identifies distinctive structural characteristics of the human CTF18–RFC complex, providing insights into its role in PCNA loading and the stimulation of leading strand synthesis by Pol ε.

## Introduction

High fidelity and efficiency are critical for chromosomal DNA duplication in eukaryotes. This precision is achieved through the coordinated action of replicative DNA polymerases, Pol δ and Pol ε, anchored to the sliding clamp, proliferating cell nuclear antigen (PCNA) (*Hubscher et al., 2002*). The ring-shaped PCNA homotrimer encircles DNA, enhancing the processivity of these polymerases in DNA replication and repair processes, including excision repair and homologous recombination (*Choe and Moldovan, 2017*), and plays an important role in sister chromatid cohesion establishment (*Moldovan et al., 2006*). The closed-ring topology of PCNA necessitates the transient opening of one interface to facilitate the incorporation and trapping of primer–template DNA. This critical function is executed by pentameric AAA+ (ATPases Associated with diverse cellular Activities) ATPases, collectively known

as clamp loaders (*Indiani and O'Donnell, 2006*; *Kelch et al., 2012*; *Kang et al., 2019*). Extensive structural and biochemical work in different organisms (*Jeruzalmi et al., 2001*; *Bowman et al., 2004*; *Kazmirski et al., 2004*; *Gaubitz et al., 2020*; *Gaubitz et al., 2022*; *Schrecker et al., 2022*) established a common mechanism for the clamp-loading reaction, which proceeds through three main steps (*Turner et al., 1999*; *Gomes et al., 2001*; *Chen et al., 2009*; *Simonetta et al., 2009*; *Kelch et al., 2011*; *Kelch, 2016*). Initially, the clamp loader engages with the intact clamp ring, adopting an autoinhibited configuration that precludes DNA binding (*Gaubitz et al., 2020*). This stage sets the groundwork for the subsequent transition, where the clamp loader alters its conformation to an activated state. This change facilitates the opening of the clamp ring, creating a gap sufficiently large to permit the passage of duplex DNA and aligning the clamp loader for primer–template DNA interaction. The final phase of this process is initiated by the binding of DNA to the clamp loader, catalysing ATP hydrolysis. DNA binding prompts the closure of the clamp and hydrolysis of ATP induces the concurrent disassembly of the closed clamp loader from the sliding clamp–DNA complex, completing the cycle necessary for the engagement of the replicative polymerases to start DNA synthesis.

The canonical replication factor C (RFC), consisting of a large RFC1 subunit and four subunits (RFC2, RFC3, RFC4, and RFC5), acts as the primary clamp loader for PCNA (*Indiani and O'Donnell, 2006*; *Jeruzalmi et al., 2002*). However, eukaryotes also employ alternative loaders (*Lee and Park, 2020*), including CTF18–RFC (*Kang et al., 2019*; *Mayer et al., 2001*; *Naiki et al., 2001*; *Bylund and Burgers, 2005*; *Crabbé et al., 2010*), which likely use a conserved loading mechanism but are functionally specialized through specific protein interactions and context-dependent roles in DNA replication.

Genome-wide analyses revealed that CTF18–RFC predominantly facilitates PCNA loading on the leading strand, in contrast to the canonical RFC complex, which mainly loads PCNA on the lagging strand (*Liu et al., 2020*). Although CTF18–RFC is not essential for bulk DNA replication in yeast (*al-Khodairy and Carr, 1992*; *Kim et al., 2005*), it plays a vital role in ensuring sister chromatid cohesion (*Mayer et al., 2001*; *Bermudez et al., 2003a*; *Terret et al., 2009*). This may be due to its selective loading of PCNA linked to Ecol acetyltransferase (*Moldovan et al., 2006*), which acetylates Smc3, preventing cohesin destabilization by Wapl (*Zhang et al., 2008*; *Rolef Ben-Shahar et al., 2008*; *Unal et al., 2008*). Recruitment of CTF18–RFC to the leading strand is also important for activation of the replication checkpoint (*Naiki et al., 2001*; *Crabbé et al., 2010*; *Kubota et al., 2013*; *Stokes et al., 2020*).

The CTF18–RFC complex features two distinct modules with separate functions (*Bylund and Burgers, 2005*; *Bermudez et al., 2003a*; *Bermudez et al., 2003b*; *Fujisawa et al., 2017*): a catalytic RFC module for PCNA loading, consisting of a large RFC1-like subunit (hereby referred to as Ctf18), alongside the four RFC2, RFC3, RFC4, and RFC5 subunits utilized by the canonical RFC complex, and a regulatory module, made up of the Ctf8 and Dcc1 subunits. These latter two subunits attach to the Ctf18 C-terminus, forming the Ctf18-1-8 module. A long linker predicted to be flexible connects these two segments (*Stokes et al., 2020*; *Grabarczyk et al., 2018*). Structural and functional studies have demonstrated that the interaction between the Ctf18-1-8 module and the catalytic domain of Pol2 (Pol2$_{CAT}$), the principal subunit of Pol ε, directs CTF18–RFC to replication forks and boosts its clamp-loading efficiency (*Stokes et al., 2020*; *Grabarczyk et al., 2018*). This interaction situates the clamp loader close to the primer/template junction, facilitating PCNA loading (*Stokes et al., 2020*). Pol2$_{CAT}$ is tethered to the remainder of Pol ε via an unstructured linker, making it an integral component of the CMGE (Cdc45-Mcm-GINS-Pol ε) core replisome complex (*Jones et al., 2021*). It has been proposed that Pol ε utilizes both the CMG helicase and PCNA as processivity factors to facilitate normal replication rates (*Langston et al., 2014*; *Yeeles et al., 2017*). Tethering by the CMG complex might allow Pol ε to dissociate from the 3′-end of the leading strand but stay at the replication fork until leading strand synthesis restarts. Given the relatively weak interaction of Pol ε with PCNA (*Chilkova et al., 2007*), it is possible that multiple PCNA loading events on the leading strand via CTF18–RFC are necessary to achieve full Pol ε processivity.

Although the structure of the Ctf18-1-8 module in association with Pol2$_{CAT}$ has been elucidated (*Stokes et al., 2020*; *Grabarczyk et al., 2018*), the architecture of the CTF18–RFC module and its interaction with PCNA remain uncharacterized. To investigate the clamp-loading mechanism employed by CTF18–RFC, we reconstituted the full human CTF18–RFC complex with PCNA and determined its structure using cryo-electron microscopy (cryo-EM). Our cryo-EM data supports the prediction that

the Ctf18-1-8 and RFC modules are flexibly tethered. The analysis yielded high-resolution reconstructions of the entire CTF18–RFC module bound to a closed PCNA ring. Although the overall architecture mirrors the autoinhibited state observed in the canonical RFC loader in a complex with PCNA (*Gaubitz et al., 2020*), distinctive features within CTF18–RFC were identified. These distinctions may offer insights into the specialized functional roles CTF18–RFC plays in DNA replication.

## Results

### Structure of the human CTF18–RFC–PCNA complex

We employed single-particle cryo-EM to determine the structure of human CTF18–RFC bound to PCNA. To this purpose, the full CTF18–RFC complex, including Ctf8 and Dcc1, was expressed in insect cells and purified to homogeneity (*Figure 1—figure supplement 1*). The purified complex includes all subunits as shown by SDS–PAGE (*Figure 1—figure supplement 1*). With the aim of trapping CTF18–RFC in different steps of the clamp-loading reaction, the complex was vitrified in the presence of ATP, PCNA, and primer/template DNA but omitting the $Mg^{2+}$ cofactor to halt hydrolysis (*Shahid et al., 2025*), and then imaged by cryo-EM (*Figure 1—figure supplement 2*). Surprisingly, image processing yielded only a major 3D class, including the CTF18–RFC pentamer bound to a closed PCNA ring (*Figure 1a, b*, *Figure 1—figure supplements 1 and 2*), with no evidence of DNA density. The Ctf18-1-8 module, which is connected to the RFC module by a 91 amino acid linker (*Figure 1a*), is not resolved in the map (*Figure 1b*), suggesting it is flexibly tethered to the RFC module. The map was refined to a global resolution of 2.9 Å (*Figure 1—figure supplements 2 and 3*), which allowed model building of the five CTF18–RFC subunits as well as the PCNA homotrimer (*Figure 1c*). The final model refines to a map-to-model FSC of 3.0 Å at 0.5 threshold, with good refinement statistics (*Supplementary file 1*).

The five CTF18–RFC subunits are positioned on top of the closed PCNA ring, which is planar and undistorted (*Figure 1c*). A structural comparison shows that the CTF18–RFC pentamer was trapped in an autoinhibited conformation analogous to that reported for the canonical human (h) and yeast (sc) RFC (*Bowman et al., 2004*; *Gaubitz et al., 2020*), pertaining to the initial step of the clamp-loading reaction, right before clamp opening (*Figure 1d*). In such conformation, the AAA+ domains form an overtwisted and asymmetric spiral that is incompatible with DNA binding (*Bowman et al., 2004*; *Gaubitz et al., 2020*). The AAA+, collar, and A′ domains of the Ctf18 subunit (*Figure 2a*) align with the homologous domains of hRFC1 (*Figure 2b*). As in the hRFC structure (*Gaubitz et al., 2020*), the Ctf18 subunit interacts with RFC2, RFC3, and PCNA-I. Three main deviations from the hRFC–PCNA structure are observed (*Gaubitz et al., 2020*): a different tilt of the PCNA ring (*Figure 1c, d*), significant mobility of the large Ctf18 subunit (*Figure 1b*, *Figure 1—figure supplement 2*) and, importantly, the presence of a β-hairpin engaged to the outer surface of the RFC5 subunit, which we mapped to the disordered N-terminus of the Ctf18 subunit (*Figure 2a, c*). These features are analysed in detail below.

### Structure of the large subunit of CTF18–RFC

The Ctf18 subunit presents 26% identity with the homologous subunit RFC1 of hRFC and an overall structural conservation, with some deviations particularly in the A′ domain (*Figure 2b*). Compared to the other subunits, the AAA+ domain of Ctf18 shows increased flexibility, as evidenced by the local resolution map of the complex as well as 3D variability analysis (*Punjani et al., 2017*; *Punjani and Fleet, 2021*; *Figure 1—figure supplement 2* and *Video 1*). Increased flexibility was not detected in the RFC1 subunit of human RFC bound to PCNA (*Gaubitz et al., 2020*). This discrepancy could be explained by diminished inter-subunit interactions involving both the AAA+ and collar domains of Ctf18 compared to RFC1, as illustrated in *Figure 1—figure supplement 4*: the AAA+ domain of Ctf18 forms fewer stabilizing salt bridges and h-bonds with RFC2 than RFC1 does, and the Ctf18 collar domain establishes markedly weaker interactions with both RFC2 and RFC3 than the extensive network observed for RFC1. Conversely, the interaction between the A′ domain of CTF18–RFC and RFC3 is strong as observed for RFC. Overall, the total buried surface area between Ctf18 and interacting subunits is comparable to that of RFC1 (~5500 vs. ~5300 Å²). However, the buried surface area between Ctf18 and RFC2 is markedly smaller than that of RFC1 (~1400 and ~2300 Å²). As explained below, the lost interactions mediated by the AAA+ and collar domains of Ctf18 are partially counterbalanced by those established with the N-terminal β-hairpin, which is absent in the canonical

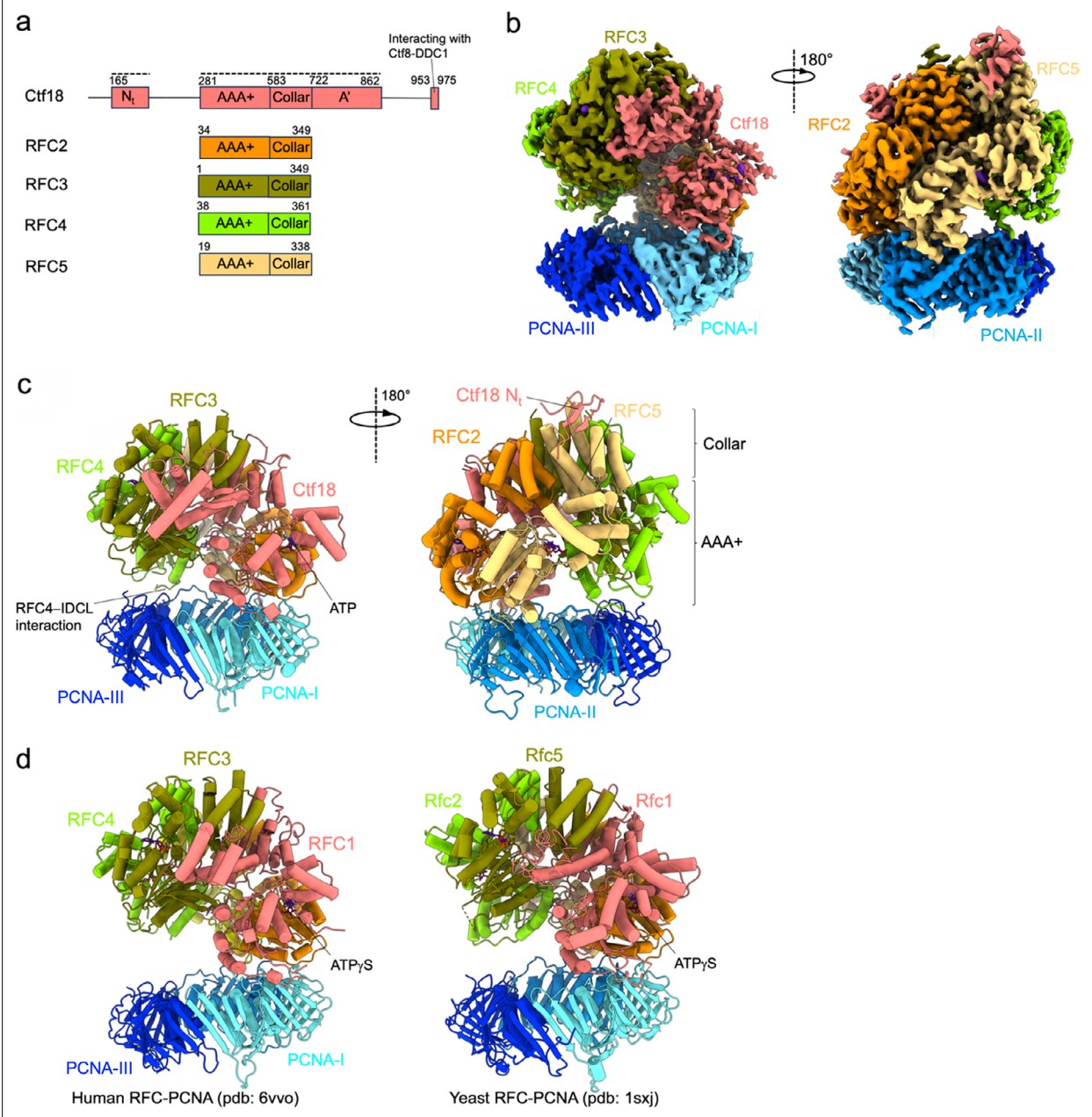

**Figure 1.** Cryo-EM structure of the human CTF18–RFC–PCNA complex in the presence of ATP. (**a**) Domain organization of the CTF18–RFC subunits forming the RFC pentamer. The dotted line above the Ctf18 subunit highlights the regions that are observed in the cryo-EM map. (**b**) Two views of the cryo-EM map of the CTF18–RFC–PCNA complex, coloured by subunits. The less sharp definition of the map region corresponding to the AAA+ domain of the Ctf18 subunit supports a relatively high mobility of this subunit. (**c**) Molecular model of the CTF18–RFC–PCNA complex in ribbon representation. (**d**) Structures of the human (left) and *Saccharomyces cerevisiae* (right) RFC–PCNA complex, which present the RFC pentamer in an autoinhibited conformation analogous to the one captured in the CTF18–RFC–PCNA complex.

The online version of this article includes the following source data and figure supplement(s) for figure 1:

**Figure supplement 1.** SDS–PAGE gel (4–20%) of the purified human clamp loader CTF18–RFC WT and CTF18$^{\Delta165–194}$–RFC.

**Figure supplement 1—source data 1.** TIFF file containing uncropped SDS–PAGE gel image indicating the relevant bands.

**Figure supplement 1—source data 2.** TIFF file containing uncropped and unlabelled SDS–PAGE gel image.

**Figure supplement 2.** Cryo-EM of the CTF18–PCNA complex in the presence of ATP.

**Figure supplement 3.** Workflow of cryo-EM image processing and 3D reconstruction of CTF18–RFC–PCNA complex in the presence of 0.5 mM ATP and without Mg$^{2+}$ (**Dataset 1**).

**Figure supplement 4.** Interfaces between RFC1 or Ctf18 and RFC2, and RFC1 or Ctf18 and RFC3.

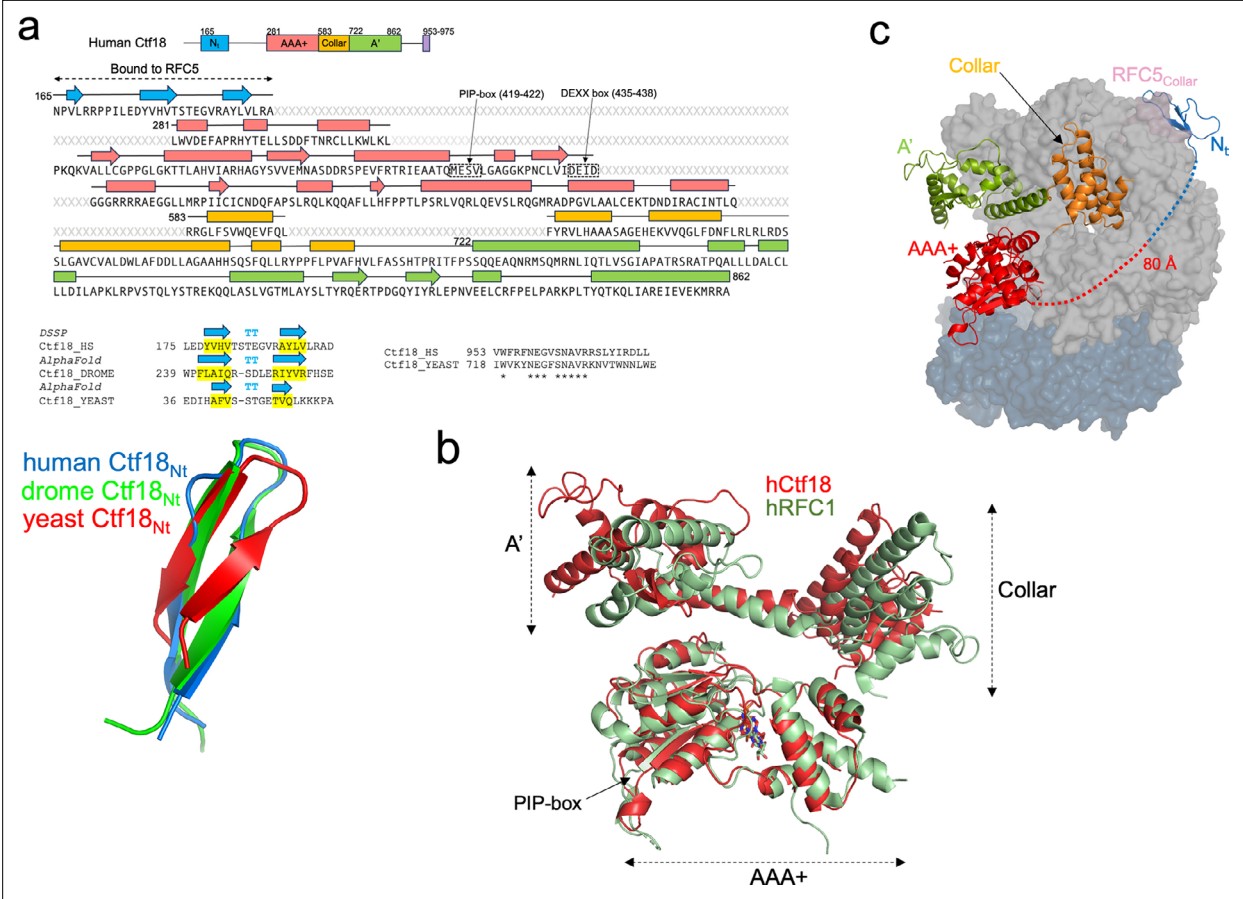

**Figure 2.** Structure of the Ctf18 subunit. (**a**) Sequence of the human Ctf18 subunit, highlighting the secondary structural elements as assigned in the cryo-EM model. The atypical PCNA-interacting motif (PIP-box) and conserved DEXX box are boxed. Residues that are not modelled in the cryo-EM structure are labelled as X in grey. Below the Ctf18 sequence to the left, the sequences of the N-terminus of human and other lower eukaryotes Ctf18, together with the secondary prediction by AlphaFold (*Jumper et al., 2021*), are shown, highlighting the conservation of the b-hairpin fold. Below, the cryo-EM β-hairpin fold model is overlaid onto the corresponding motifs predicted by AlphaFold for *Drosophila* or *Saccharomyces cerevisiae* Ctf18. Below the Ctf18 sequence to the right, the alignment of the C-terminal region of human and *S. cerevisiae* Ctf18, which interacts with the Ctf8/Dcc1 subunits, is shown. Asterisks denote conserved residues. (**b**) Superposition of the human Ctf18 and RFC1 (PDB: 6VVO) subunit structures, with subdomains labelled. (**c**) Structure of the Ctf18 subunit, in ribbon representation, within the CTF18–RFC–PCNA complex. The Ctf18 subdomains are colour-coded as in panel (**a**). The other complex subunits are shown as surfaces.

RFC. Additionally, the interaction between Ctf18 and PCNA is weaker compared to the interaction between RFC1 and PCNA, further contributing to these structural differences (see next section). The Ctf18 C-terminal region (residues 953–975) predicted to interact with the Ctf8 and Dcc1 subunits (*Figure 2a*) is invisible in the map.

The most striking feature is a small β-hairpin-containing fold mapped between residues 165–194 in the disordered N-terminus (*Figure 2a, c*) of the Ctf18 subunit, ~80 Å apart from the Ctf18 AAA+ domain, to which it is connected by a disordered linker of 86 amino acids (residues 195–280). This N-terminal fold plugs into a groove at the outer surface of the collar domain of the RFC5 subunit defined by helices α1, α3, α4, and α5 (*Figure 3a*). The resolution of the map in this region is sufficient for confident model building of the interface (*Figure 3b*). The interface buries 984 Å² and is characterized by both apolar and polar interactions (*Figure 3a*). A set of hydrophobic side chains protruding from Ctf18 β-hairpin (V178, V180, L190, and L192) approaches a complementary patch of hydrophobic residues on RFC5 (L255 in α1; F287, V291, and F293 in α3; I336 and V337 in α5). The interface is further strengthened by several hydrogen bonds mostly mediated by main-chain atoms in the Ctf18 β-hairpin, and by a salt bridge between R193 of the Ctf18 β-hairpin and D292 of RFC5. It is worth highlighting that RFC2 separates the Ctf18 and RFC5 subunits. Despite this separation,

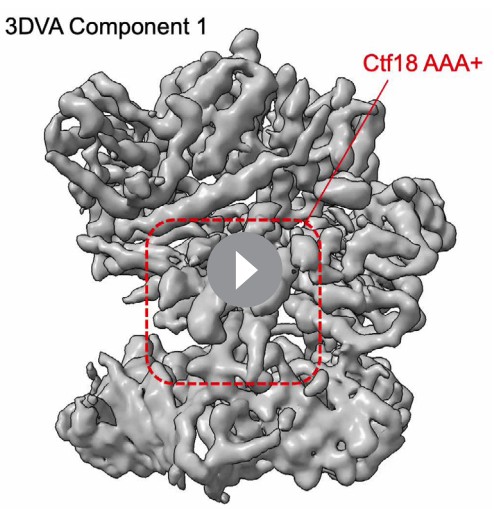

3DVA Component 1

Ctf18 AAA+

**Video 1.** 3DVA analysis of the human CTF18–RFC–PCNA complex in the presence of ATP. The video shows morphing transitions across the first three principal components obtained by 3D variability analysis (3DVA). Three volumes per component are morphed, illustrating continuous conformational flexibility within the complex. The dominant motion is localized to the AAA+ domain of the CTF18 subunit, consistent with reduced local resolution in this region and supporting a model in which this domain.

https://elifesciences.org/articles/103493/figures#video1

Ctf18 projects the β-hairpin directly onto RFC5, bypassing any interactions with RFC2 (*Figure 2c*). This observation supports that the β-hairpin functions as a 'latch' to stabilize the CTF18–RFC structure.

Similarly to human CTF18–RFC, long regions predicted to be disordered by AlphaFold (*Jumper et al., 2021*) are present in the N-terminal sequence of Ctf18 of lower eukaryotes. The β-hairpin region of human Ctf18 does not show strict sequence conservation, but a similar fold in the N-terminus of lower eukaryotes Ctf18 is predicted (*Figure 2a*). Therefore, lower eukaryotes may use a similar β-hairpin motif to bind the corresponding subunit of the RFC–module complex (RFC5 in human, Rfc3 in *S. cerevisiae*), emphasizing its importance.

## Interactions of CTF18–RFC with PCNA

CTF18–RFC engages two of the three PCNA protomers through four contact points involving the AAA+ domains of Ctf18, RFC2, RFC4, and RFC5 (*Figures 1c and 3c*). The interactions of PCNA with the RFC2 and RFC5 subunits are analogous to those reported in the hRFC–PCNA complex (*Gaubitz et al., 2020*). However, while Ctf18 engages PCNA-I via a PIP-box interaction similar to hRFC1 (*Gaubitz et al., 2020*), its atypical PIP motif—lacking aromatic residues

(*Figures 2a and 3c*)—results in a smaller buried surface area (~780 vs. ~1010 Å²). A methionine residue (M419) inserts into the so-called 'Q-pocket', which is typically occupied by a glutamine in canonical PIP-boxes, while V422 occupies the conserved hydrophobic pocket that usually accommodates aromatic residues. This low-affinity interaction does not stably anchor Ctf18 to PCNA-I and may contribute to the relative flexibility of the AAA+ domain of Ctf18 observed in the complex (*Figure 1—figure supplement 2* and *Video 1*). The PIP-box of RFC5 is also atypical but contains two hydrophobic residues (I114 and F115) that insert into the conserved pocket of PCNA-II. The interacting motifs of RFC2 and RFC4 differ from each other in sequence but both engage similar regions involving the interdomain connecting loop (IDCL) of PCNA-I and PCNA-II, respectively (*Figure 3d*).

Interestingly, while RFC4 interacts with PCNA in the CTF18–RFC complex, it does not in the hRFC–PCNA complex (*Gaubitz et al., 2020*). The RFC4–PCNA interaction, which was also observed in the structure of *S. cerevisiae* RFC bound to open PCNA (*Gaubitz et al., 2022*), brings PCNA closer to the CTF18–RFC module, thereby explaining the distinct tilt of the PCNA ring observed in our structure compared to the hRFC–PCNA complex (*Figure 1c, d*). Given the structure of scRFC bound to open PCNA (*Gaubitz et al., 2022*), where all five RFC subunits are bound to PCNA (*Figure 3d*), we predict that a conformational shift in CTF18–RFC, specifically bringing the fifth subunit (RFC3) to engage PCNA-III, would be necessary to disrupt the interface between PCNA-I and PCNA-III, thereby opening the clamp and aligning the CTF18–RFC subunits for primer–template DNA interaction. The reason for the absence of such an active CTF18–RFC conformation in our cryo-EM analysis remains uncertain. However, similar to observations with the hRFC–PCNA complex, where an open-clamp state was not reported (*Gaubitz et al., 2020*), this could be attributed to the hydrophobic regions of the open clamp interacting with the water–air interface, potentially leading to protein denaturation during the vitrification process to prepare the cryo-EM grids.

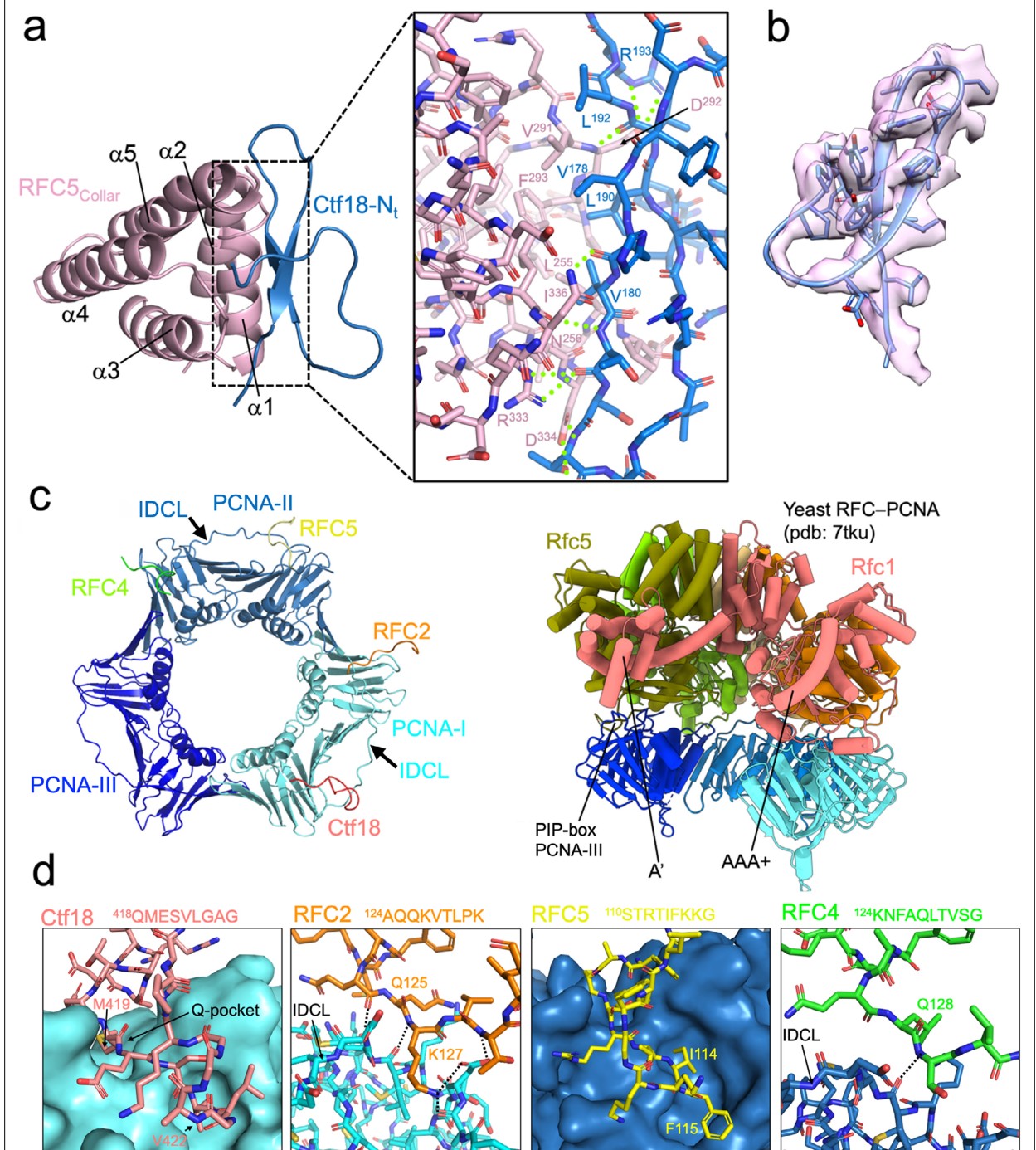

**Figure 3.** Structural details of the Ctf18–RFC5 and CTF18–RFC–PCNA interactions. (**a**) Model of the RFC5 collar domain and Ctf18 N-terminal b-hairpin in ribbon representation. The inset shows the interface in stick representation, with hydrogen bonds shown as green dotted lines. (**b**) Map region around the Ctf18 N-terminal b-hairpin. (**c**) Model of the PCNA homotrimer bound to the four CTF18 interacting regions. To the right, the structure of the yeast RFC–PCNA complex with an open clamp, highlighting the engagement of all five RFC subunits to PCNA, required for clamp opening. (**d**) Model of the regions of the CTF18–RFC–PCNA interactions, with CTF18–RFC subunit residues shown as sticks and PCNA as surface or ribbon. Polar interactions are shown as black dotted lines. The amino acid sequence encompassing the interacting motifs in the various subunits is shown.

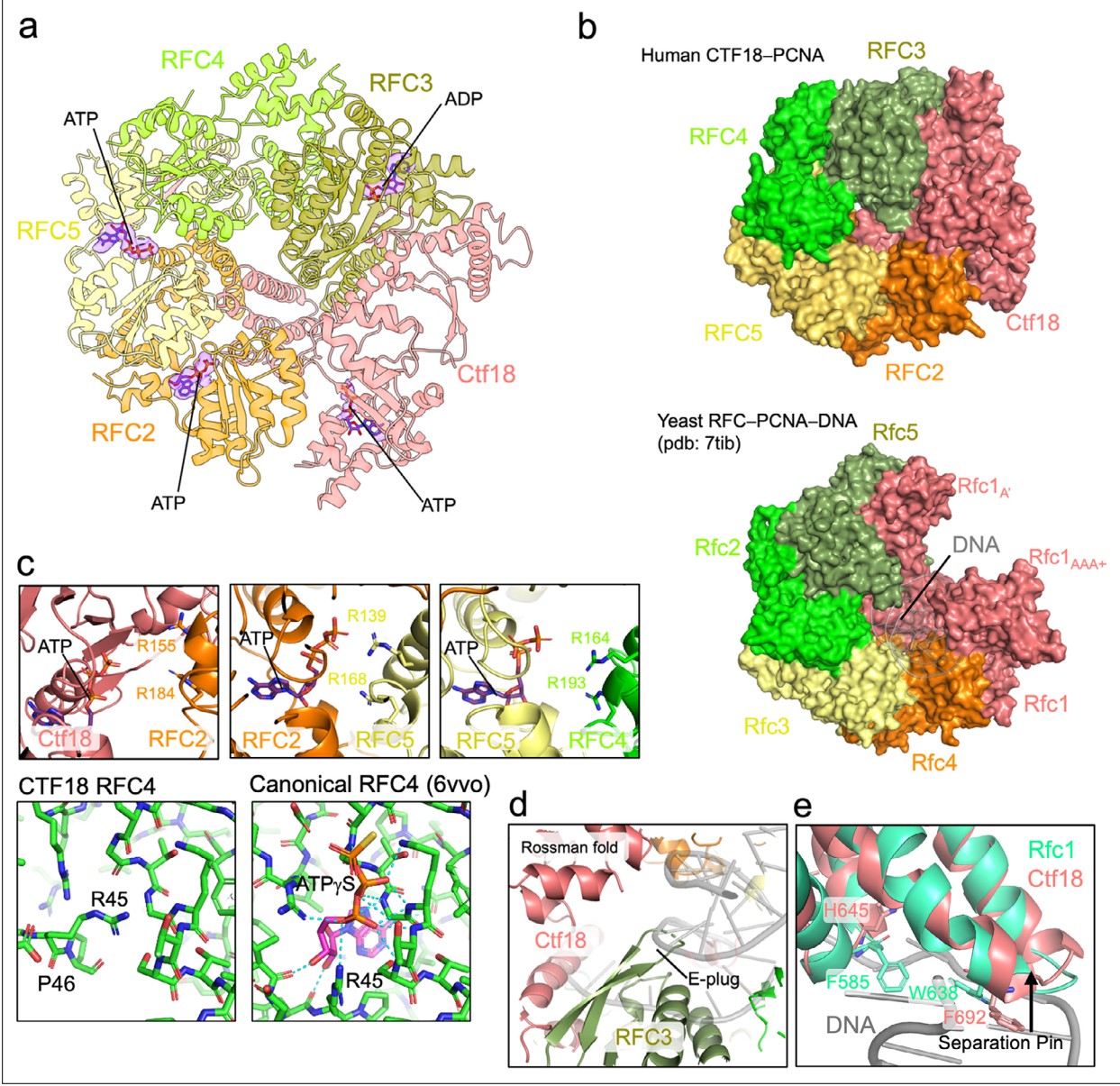

**Figure 4.** Analysis of the autoinhibited conformation of the CTF18 pentamer and nucleotide coordination. (**a**) Ribbon model of the CTF18–RFC pentamer (PCNA hidden). Nucleotide molecules are shown as sticks, with the corresponding cryo-EM density. (**b**) Surface representation of (top) the CTF18–RFC pentamer and (bottom) the yeast RFC pentamer bound to open PCNA and primer–template DNA. PCNA is hidden in both structures. The overtwisted CTF18–RFC pentamer cannot accommodate DNA within the inner chamber. (**c**) Model of the ATP active sites in the CTF18–RFC–PCNA complex, showing disruption of the interactions mediated by the arginine fingers. Below, insets show the nucleotide-binding site of subunit RFC4 in either the human CTF18–RFC (left) or human RFC (right) models, highlighting the structural incompatibility with nucleotide binding in the CTF18–RFC structure. Polar interactions with ATPgS in human RFC4 are indicated as cyan dotted lines. (**d**) Detail of the inner chamber region of the CTF18–RFC pentamer, showing the b-hairpin (E-plug) of RFC5 protruding in the region predicted to be occupied by DNA. The DNA (grey ribbon) is modelled as in the yeast RFC–DNA–PCNA complex. (**e**) The structures of the RFC1 subunit from the yeast RFC–DNA–PCNA complex (*Gaubitz et al., 2022*) and the Ctf18 subunit from the CTF18–RFC–PCNA complex are aligned to highlight conservation of the separation pin. Side chains of residues implicated in this motif in yeast RFC, along with the corresponding amino acids in CTF18–RFC, are shown.

The online version of this article includes the following figure supplement(s) for figure 4:

**Figure supplement 1.** Walker A sequence and structure.

## Nucleotide binding to the AAA+ modules

The map is consistent with nucleotides in four of the five AAA+ domains of the CTF18–RFC subunits (*Figure 4a*). ATP is bound to subunits Ctf18, RFC2, and RFC5, while ADP appears to engage RFC3, consistent with the structure of the hRFC–PCNA complex reconstituted with the slowly hydrolysable nucleotide ATPγS (*Gaubitz et al., 2020*). Density for the adenine ring of ATP in the Ctf18 subunit is poor, likely a consequence of the increased mobility of the AAA+ domain of this subunit (*Figure 4a*). Compared to the canonical RFC1, the Walker A motif of Ctf18 contains a leucine residue (L378) instead of a valine, but this substitution does not appear to affect its interaction with the ATP nucleotide (*Figure 4—figure supplement 1*). Since RFC3 lacks critical catalytic residues and bears a substitution in the Walker A motif (*Gaubitz et al., 2020*), the presence of ADP, rather than a product of ATP hydrolysis, probably results from co-purification of the complex where ADP contaminants are present. In the human RFC complex, it has been previously shown that an intact nucleotide binding site in RFC3 has an important structural role in the complex assembly (*Cai et al., 1996*; *Podust et al., 1998*). If this is the case for CTF18–RFC, it remains to be established, but it is likely that ADP binding to this subunit may play a stabilizing role in all RFC-like loaders.

Differently from the hRFC–PCNA complex, where RFC4 is bound to ATPγS, in CTF18–RFC, this subunit is not engaged with a nucleotide (*Figure 4a*). The residues in the AAA+ of RFC4 of CTF18–RFC coordinated to ATPγS in hRFC are displaced and incompatible with nucleotide binding (*Figure 4c*). In particular, the side chains of R45 and P46 in RFC4 occupy the space where ATP would normally bind, thereby sterically hindering nucleotide accommodation in the CTF18–RFC structure (*Figure 4c*). Interestingly, mutation of the conserved K84 residue in the ATP-binding motif of RFC4 in hRFC did not impair the ability of this subunit to assemble with other RFC subunits (*Cai et al., 1996*; *Podust et al., 1998*). In agreement, our structure shows that the lack of ATP binding to RFC4 does not preclude the assembly of the CTF18–RFC pentamer in the autoinhibited conformation. However, considering the importance of ATP binding of RFC4 for the hRFC clamp-loading function (*Cai et al., 1998*; *Schmidt et al., 2001*; *Seybert and Wigley, 2004*; *Sakato et al., 2012*), it is expected that ATP binding to RFC4 in CTF18–RFC is required to switch to an enzymatically active conformation.

The asymmetric spiral of the CTF18–RFC AAA+ pentamer (*Figure 4a, b*) disrupts the active sites at the Ctf18/RFC2, RFC2/5, and RFC5/4 interfaces (*Figure 4c*). At these interfaces, the *trans*-acting arginine fingers are positioned too far away to interact with the γ-phosphate of ATP (e.g., ~9 Å for R184 of RFC2, ~10 Å for R168 of RFC5, and ~7 Å for R193 of RFC4) (*Figure 4c*). Prior research has indicated that hydrolysis at the large subunit/RFC2 interface is not essential for clamp loading by various loaders (*Cai et al., 1998*; *Schmidt et al., 2001*; *Seybert and Wigley, 2004*; *Sakato et al., 2012*), while the others are critical for the clamp-loading activity of eukaryotic RFCs.

In CTF18–RFC, the AAA+ spiral is overtwisted in a manner reminiscent of the autoinhibited conformations observed in hRFC and scRFC (*Bowman et al., 2004*; *Gaubitz et al., 2020*; *Figure 1d*), leading to steric hindrance that impedes DNA binding (*Figure 4b*). Specifically, the overtwisted RFC3 and RFC4 subunits occupy the space normally filled by DNA. In the autoinhibited structure of hRFC, the β-hairpin (residues 68–88) located at the base of RFC3, also referred to as 'E-plug', extends towards the anticipated location of DNA and interacts with the Rossmann fold of RFC1 (*Gaubitz et al., 2020*). Conversely, in CTF18–RFC, while this same β-hairpin is oriented towards the interior of the DNA-binding channel (*Figure 4d*), it does not make direct contact with the Ctf18 subunit. Collectively, the E-plug appears a conserved feature in eukaryotic clamp loaders, underlying its importance in maintaining the pentamer in its autoinhibited conformation, by preventing DNA access through the central channel. We observed that human CTF18–RFC shares an additional structural feature with both human and yeast RFC: the presence of a separation pin in the large subunit (*Gaubitz et al., 2020*; *Gaubitz et al., 2022*). Although the separation pin is not essential for clamp loading (*Gaubitz et al., 2022*), its conservation across species and clamp loader types suggests an auxiliary or context-dependent function. Interestingly, residues W638 and F582—proposed to stabilize the separation pin and its interaction with DNA in yeast RFC—are substituted by F692 and H645 in human CTF18–RFC, respectively (*Figure 4e*), possibly indicating a mechanistically similar role in both systems.

To discount the possibility that the absence of the $Mg^{2+}$ ions in the cryo-EM sample could have hindered the proper assembly of the CTF18–RFC pentamer for DNA engagement, we used cryo-EM to visualize CTF18–RFC with PCNA, primer–template DNA, ATP, and $Mg^{2+}$ (*Figure 5—figure supplements 1 and 2*; *Supplementary file 1*). Despite the inclusion of $Mg^{2+}$, the resulting 3.2 Å reconstruction

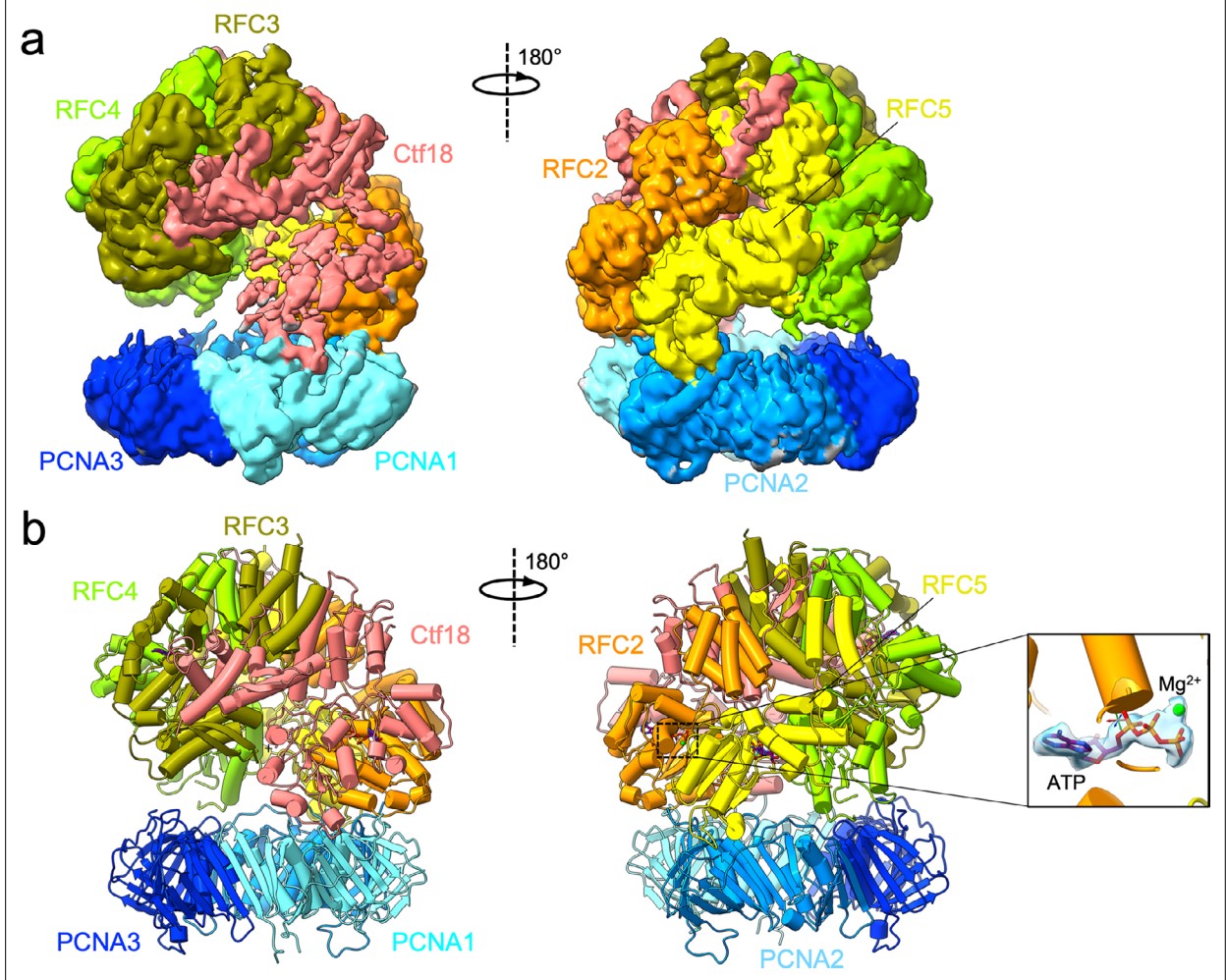

**Figure 5.** Cryo-EM structure of the human CTF18–RFC–PCNA complex in the presence of ATP and Mg²⁺. (**a**) Two views of the cryo-EM map, coloured by subunits. (**b**) Molecular model in ribbon representation. The inset shows a map and model of the ATP molecule bound to the RFC2 subunit, with a discernible Mg²⁺ ion.

The online version of this article includes the following figure supplement(s) for figure 5:

**Figure supplement 1.** Cryo-EM of the CTF18–PCNA complex in the presence of ATP and Mg²⁺.

**Figure supplement 2.** Workflow of cryo-EM image processing and 3D reconstruction of CTF18–RFC–PCNA complex in the presence of 0.5 mM ATP and 5 mM Mg²⁺ (**Dataset 2**).

of the complex remained in a configuration analogous to that without Mg²⁺ (*Figure 5a, b*, *Figure 5— figure supplement 1*). In this reconstruction, distinct density for Mg²⁺ is noted in the active sites of RFC2 and RFC5, coordinating ATP (*Figure 5b*). Similar to the map without Mg²⁺, ADP coordination is present in RFC3, while RFC4 is not engaged with any nucleotide. These findings collectively imply that the presence of Mg²⁺ in the nucleotide binding sites does not influence the transition of CTF18–RFC from an autoinhibited to an active state. This result also confirms the remarkable stability of the auto-inhibited conformation, even in hydrolysing conditions.

## The Ctf18 β-hairpin plays a role in PCNA loading and in stimulating DNA synthesis by Pol ε

Based on the structures reported herein, the β-hairpin at the disordered N-terminal domain of Ctf18 is a distinctive feature that may serve to stabilize the CTF18–RFC pentamer and maximize its clamp-loading efficiency. To investigate the functional importance of the β-hairpin, we purified a CTF18–RFC mutant with a truncation at the disordered N-terminal domain of the Ctf18 subunit,

specifically, the segment spanning residues 165–194 (CTF18$^{\Delta165–194}$–RFC). This mutation eliminates the entire Ctf18 β-hairpin and is expected to destabilize the CTF18–RFC pentamer. In agreement, when mutant CTF18–RFC was imaged by cryo-EM with PCNA in the same conditions used for wild-type CTF18–RFC, only a small subset of particles of the complex was observed, while the majority of particles contained dissociated PCNA (*Figure 6—figure supplements 1 and 2*). We hypothesized that this destabilizing mutation in CTF18–RFC may reduce its efficiency in loading PCNA onto DNA.

We directly compared clamp loading by CTF18–RFC, CTF18$^{\Delta165–194}$–RFC and the canonical RFC complex, using a pre-steady-state ensemble FRET assay to monitor the loading of Cy5-labelled PCNA onto a Cy3-labelled primer–template DNA substrate pre-coated with RPA (*Figure 6a, b*, *Figure 6— figure supplement 3–7*). Prior to the addition of clamp loaders (*Figure 6—figure supplement 4a*), donor ($I_{563}$) and acceptor ($I_{665}$) fluorescence intensities remained low and stable over time (*Figure 6— figure supplement 4b–d*). Here, $I_{563}$ and $I_{665}$ correspond to the fluorescence intensities of the Cy3 donor (563 nm) and Cy5 acceptor (665 nm), respectively, and were used to calculate the apparent FRET efficiency ($E_{FRET}$). The constant $E_{FRET}$ values recorded during the first 60 s represent the experimental baseline ($Y_{Min}$) corresponding to the absence of interactions between Cy5–PCNA and the 5'ddPCy3/T·RPA nucleoprotein complex (*Norris et al., 2024*; *Norris and Hedglin, 2024*). Upon addition of a clamp loader, $I_{665}$ increased concomitantly with a decrease in $I_{563}$, after which both signals stabilized, indicating the onset and persistence of FRET (*Figure 6—figure supplement 4b–d*). These synchronized, anti-correlated changes in donor and acceptor intensities report the loading of Cy5– PCNA onto the DNA substrate. After reaching a plateau, $E_{FRET}$ values remained constant for at least 60 s, representing the amount of loaded and stably bound PCNA rings (*Norris et al., 2024*). For each condition, $E_{FRET}$ values within the final plateau region were fit to flat lines to determine the maximum $E_{FRET}$ ($Y_{Max}$) reached after clamp loader addition. Importantly, CTF18–RFC and CTF18$^{\Delta165–194}$– RFC displayed similar dynamic ranges of $E_{FRET}$ values (i.e., $Y_{Max} − Y_{Min}$, *Figure 6—figure supplement 4c, d*), indicating that the β-hairpin deletion does not affect the overall extent of PCNA loading. All $E_{FRET}$ values measured after addition of each clamp loader were normalized to their respective dynamic ranges, defined by the baseline ($Y_{Min}$) and maximum ($Y_{Max}$) $E_{FRET}$ values, and plotted as a function of time following complex addition.

Under our experimental conditions, the clamp loader complexes must bind ATP and free PCNA before initiating loading, which can give rise to multiple kinetic phases. Such multiphasic behaviour has been described previously for RFC in stopped-flow experiments and is also evident in our ensemble FRET setup (*Norris et al., 2024*; *Norris and Hedglin, 2024*; *Hedglin and Benkovic, 2017*). The kinetic steps represented by each phase have been well defined for RFC but not for CTF18–RFC. In our measurements, the instrumental dead time was <7 s. Within this interval, a substantial fraction of PCNA loading (ranging from ~30% for CTF18–RFC to ~70% for RFC; *Figure 6b–d*) occurs, indicating clear kinetic differences among the clamp loader complexes but precluding full resolution of all reaction phases for CTF18–RFC.

To enable direct comparison of loading rates, only normalized $E_{FRET}$ values recorded after the dead time were fitted to the minimal kinetic model that yielded randomly distributed residuals. For CTF18– RFC, the data were best described by a single-exponential rise (*Figure 6—figure supplement 5*), whereas RFC and CTF18$^{\Delta165–194}$–RFC required a double-exponential fit (*Figure 6—figure supplement 6a, b*), consistent with the presence of an additional kinetic phase (*Figure 6—figure supplement 6c, d*). From these minimal models, we calculated the times required to reach 50%, 90%, and 95% of the maximal normalized $E_{FRET}$ signal ($t_{0.50}$, $t_{0.90}$, and $t_{0.95}$), corresponding, respectively, to half completion and near completion of PCNA loading (*Figure 6b–d*, *Figure 6—figure supplements 6 and 7*). For RFC, $t_{0.50}$ occurred within the dead time and is thus reported as a lower limit (*Figure 6b*).

Across the three complexes, the kinetics of PCNA loading differed significantly (*Figure 6b–d*). RFC reached half completion within the experimental dead time ($t_{0.50} \leq 7$ s), consistent with rapid loading, whereas CTF18–RFC and CTF18$^{\Delta165–194}$–RFC exhibited slower half-times of approximately 8 and 12 s, respectively. Completion times further underscored these differences: both RFC and CTF18–RFC reached 95% completion at ~77 s, while the CTF18$^{\Delta165–194}$–RFC mutant required ~145 s. This indicates that the β-hairpin deletion significantly slows the reaction progression. Altogether, the results from these FRET studies indicate that deletion of the β-hairpin from CTF18 does not affect the overall extent of PCNA loading but rather slows the reaction progression.

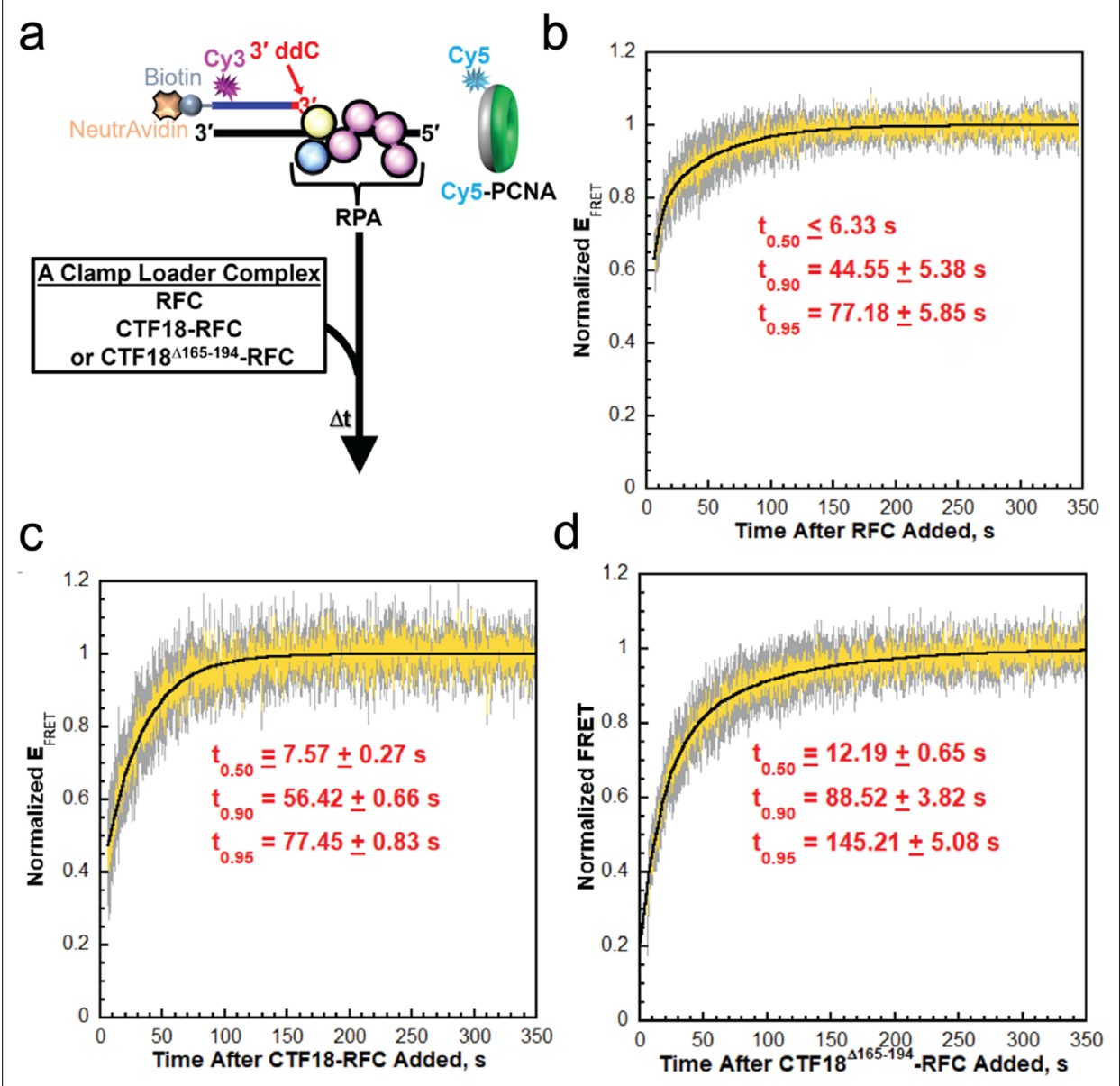

**Figure 6.** FRET assays to monitor loading of PCNA onto P/T junctions by human clamp loader complexes. (**a**) Schematic representation of the FRET pair and experiment to monitor loading of Cy5-PCNA onto 5′ddPCy3/T DNA substrates engaged by RPA. The front and back faces of PCNA are displayed in green and grey, respectively. When loaded onto a 5′ddPCy3/T DNA substrate, the Cy5 label on the back face of PCNA is oriented towards the Cy3 label near the blunt duplex end of the 5′ddPCy3/T DNA substrate, yielding a FRET signal (**b–d**) data. Each normalized $E_{FRET}$ trace is the mean of at least three independent traces with the SEM shown in grey. Each respective trace is fit to its corresponding minimal kinetic model and the times required for the trace to reach 50% ($t_{0.50}$), 90% ($t_{0.90}$), and 95% ($t_{0.95}$) of the maximal value (i.e., 1.0) are reported. Data for loading of PCNA by RFC, CTF18, and CTF18$^{\Delta165–194}$–RFC are displayed in panels **b**, **c**, and **d**, respectively.

The online version of this article includes the following figure supplement(s) for figure 6:

**Figure supplement 1.** Workflow of cryo-EM image processing and 3D reconstruction of *CTF18–RFC$^{\Delta165–194}$–PCNA* complex in the presence of 0.5 mM ATP and without Mg$^{2+}$ (Dataset 3).

**Figure supplement 2.** 2D classification of CTF18 WT and CTF18$^{\Delta165–194}$–RFC.

**Figure supplement 3.** P/T DNA substrate utilized in the FRET studies.

**Figure supplement 4.** FRET assays to monitor loading of PCNA onto P/T junctions by human clamp loader complexes.

**Figure supplement 5.** The minimal kinetic model for the increase observed in the normalized $E_{FRET}$ trace for CTF18 is a single-exponential rise.

*Figure 6 continued on next page*

**Figure supplement 6.** The minimal kinetic models for the increase observed in the normalized $E_{FRET}$ traces for RFC and CTF18$^{\Delta165-194}$–RFC are double-exponential rises.

**Figure supplement 7.** Direct comparisons of the minimal kinetic models for the increase observed in the normalized $E_{FRET}$ traces for RFC (black), CTF18–RFC (green), and CTF18$^{\Delta165-194}$–RFC (purple).

We then tested whether the observed loading defect observed with CTF18$^{\Delta165-194}$–RFC results in diminished stimulation of primer synthesis by Pol ε. To test this hypothesis, we performed primer extension assays using Pol ε, PCNA, and either wild-type or mutant CTF18–RFC constructs to assess their ability to stimulate DNA synthesis by Pol ε. Our results indicate a significant reduction in primer extension products when the β-hairpin was deleted (***Figure 7a***). Measurement of band intensities revealed an approximately twofold decrease in the amount of extended primer produced by the mutant compared to the wild-type (***Figure 7b***). To further support that the observed effect on primer extension is due to the mutation, we conducted a concentration titration comparing wild-type and mutant CTF18–RFC (***Figure 7—figure supplement 1***). The CTF18–RFC mutant displayed altered concentration dependencies compared to the wild-type, thereby validating the mutant phenotype and underscoring the structural importance of the β-hairpin feature. Notably, the primer extension assay was conducted with 2 nM plasmid DNA and excess CTF18–RFC (40 nM), PCNA (100 nM), and Pol ε (30 nM), establishing single-turnover conditions per DNA molecule. Under these conditions, the rate—not the extent—of PCNA loading becomes the key determinant of synthesis dynamics. Although Pol ε can initiate synthesis independently of PCNA, the clamp is required for full processivity. Therefore, the slower loading rate observed for the CTF18–RFC mutant likely delays formation of

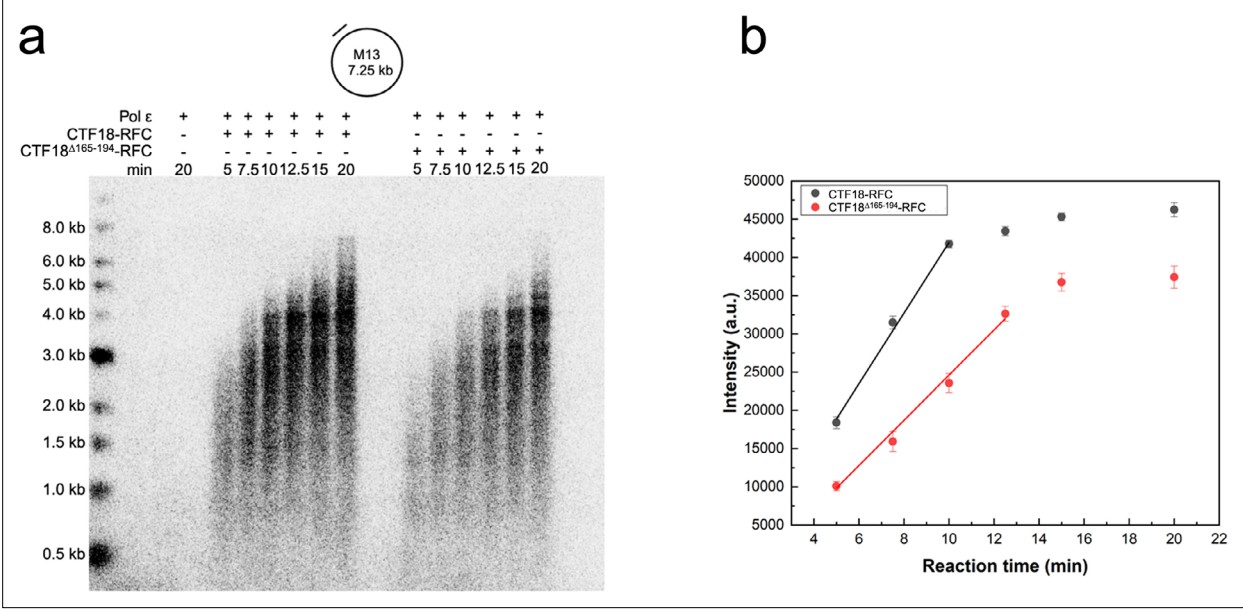

**Figure 7.** Primer extension assay with Pol ε and CTF18–RFC WT or CTF18$^{\Delta165-194}$–RFC. (**a**) Time course reactions on M13mp18 single-strand DNA with 40 nM wild-type CTF18–RFC or CTF18$^{\Delta165-194}$–RFC. Reactions were performed as described in Materials and methods. (**b**) Quantification of the band intensities from the primer extension assay shows that the rate of increase in intensity, derived from the initial linear portion of the reaction time before saturation, is 4142 ± 95 a.u./min for CTF18–RFC and 2389 ± 143 a.u./min for CTF18$^{\Delta165-194}$–RFC. This indicates that the mutant slows Pol ε synthesis by 42%. The experiment was conducted three times. Dots in (**b**) represent mean values from the three replicas.

The online version of this article includes the following source data and figure supplement(s) for figure 7:

**Source data 1.** TIFF file containing uncropped agarose gel image indicating the relevant bands.

**Source data 2.** TIFF file containing uncropped and unlabelled agarose gel image.

**Figure supplement 1.** Primer extension assays with Pol ε and CTF18 WT and CTF18$^{\Delta165-194}$–RFC.

**Figure supplement 1—source data 1.** TIFF file containing uncropped agarose gel image indicating the relevant bands.

**Figure supplement 1—source data 2.** TIFF file containing uncropped and unlabelled agarose gel image.

highly processive complexes, resulting in a slower accumulation of extended products and explaining the observed ~2-fold reduction in synthesis.

## Discussion

In this work, we reported the structure of the human CTF18–RFC clamp loader bound to PCNA in the presence of ATP with or without $Mg^{2+}$ ions. Our cryo-EM data, together with previous work (*Stokes et al., 2020*; *Grabarczyk et al., 2018*), support that the Ctf8 and Dcc1 subunits of CTF18–RFC, forming the regulatory module interacting with Pol ε (*Grabarczyk et al., 2018*), are flexibly tethered to the RFC–module via binding to the Ctf18 C-terminus, which is separated by the N-terminal RFC1-like module of Ctf18 by ~90 amino acid flexible linker. Therefore, CTF18–RFC comprises two structurally distinct modules that cooperate to load PCNA onto the leading strand, facilitating replication by Pol ε.

The CTF18–RFC–PCNA complex was observed in an autoinhibited conformation, which prevents DNA binding. This mirrors the similar conformation in hRFC and scRFC, as previously reported (*Gaubitz et al., 2020*; *Gaubitz et al., 2022*). Such conformation represents a transient encounter complex that occurs early in the clamp-loading process, before the opening of the clamp. Despite the variations in the large RFC subunit in the two assemblies, we demonstrate a striking architectural conservation between the alternative human clamp loader CTF18–RFC and the canonical RFC. Specifically, CTF18–RFC utilizes four subunits (Ctf18, RFC2, RFC5, and RFC4) to engage two PCNA protomers. However, the engagement of the third PCNA protomer by RFC3 is necessary to initiate the opening of the PCNA ring interface, a process that requires a conformational shift of the AAA+ pentamer from an autoinhibited to an active state. Supporting this, previous structural studies on scRFC have shown that the conformation of RFC bound to open PCNA is facilitated through a 'crab-claw' mechanism, which is inherently predisposed to DNA binding (*Gaubitz et al., 2022*). This conformational change creates an opening between the A' segment and the AAA+ module in the A subunit, allowing primer–template DNA to bind within the RFC inner chamber (*Gaubitz et al., 2022*; *Figure 4c*). Remarkably, this significant conformational shift occurs rapidly (*Liu et al., 2017*), does not depend on ATP hydrolysis, and ensures the sequential binding of PCNA and the DNA substrate in the correct order. While our study did not resolve the active conformation of CTF18–RFC, the observed structural conservation in the autoinhibited state, particularly of the large Ctf18 subunit (*Figure 2*), suggests that a similar 'crab-claw' mechanism may facilitate the transition of CTF18–RFC to its active state for PCNA opening and DNA binding. While our work was being finalized, several cryo-EM structures of human CTF18–RFC bound to PCNA and primer/template DNA were reported by another group (*He et al., 2024*). These findings are consistent with the distinct features of CTF18–RFC observed in our structures and independently support the notion of significant mechanistic similarity between CTF18–RFC and canonical RFC in loading PCNA onto a ss/dsDNA junction.

The main structural differences between CTF18–RFC and the canonical RFC include increased mobility of the large subunit and the presence of a β-hairpin that attaches to the RFC5 subunit. The mobility of the large subunit is likely enhanced by the low-affinity PIP motif that anchors it to the PCNA ring. We propose that the β-hairpin may have evolved to counterbalance the mobility of the large subunit, helping to maintain its connection to the RFC pentamer and stabilizing the clamp loader structure. Collectively, our structural and biochemical analyses reveal that removal of the β-hairpin compromises the stability of the CTF18–RFC–PCNA complex, results in slower clamp-loading kinetics, and leads to reduced stimulation of Pol ε primer synthesis, consistent with a defect in the rate of PCNA loading (*Figure 8*). Interestingly, a recent work reporting cryo-EM structures of the yeast homolog of CTF18–RFC bound to PCNA (*He et al., 2024*) does not support the presence of a stable β-hairpin at the N-terminus of the Ctf18 subunit engaged to the collar of the Rfc3 subunit. Therefore, the functional β-hairpin we observed in human CTF18–RFC appears a distinct feature that pertains to higher eukaryotes, required to maximize the activity of CTF18–RFC in loading PCNA onto the leading strand.

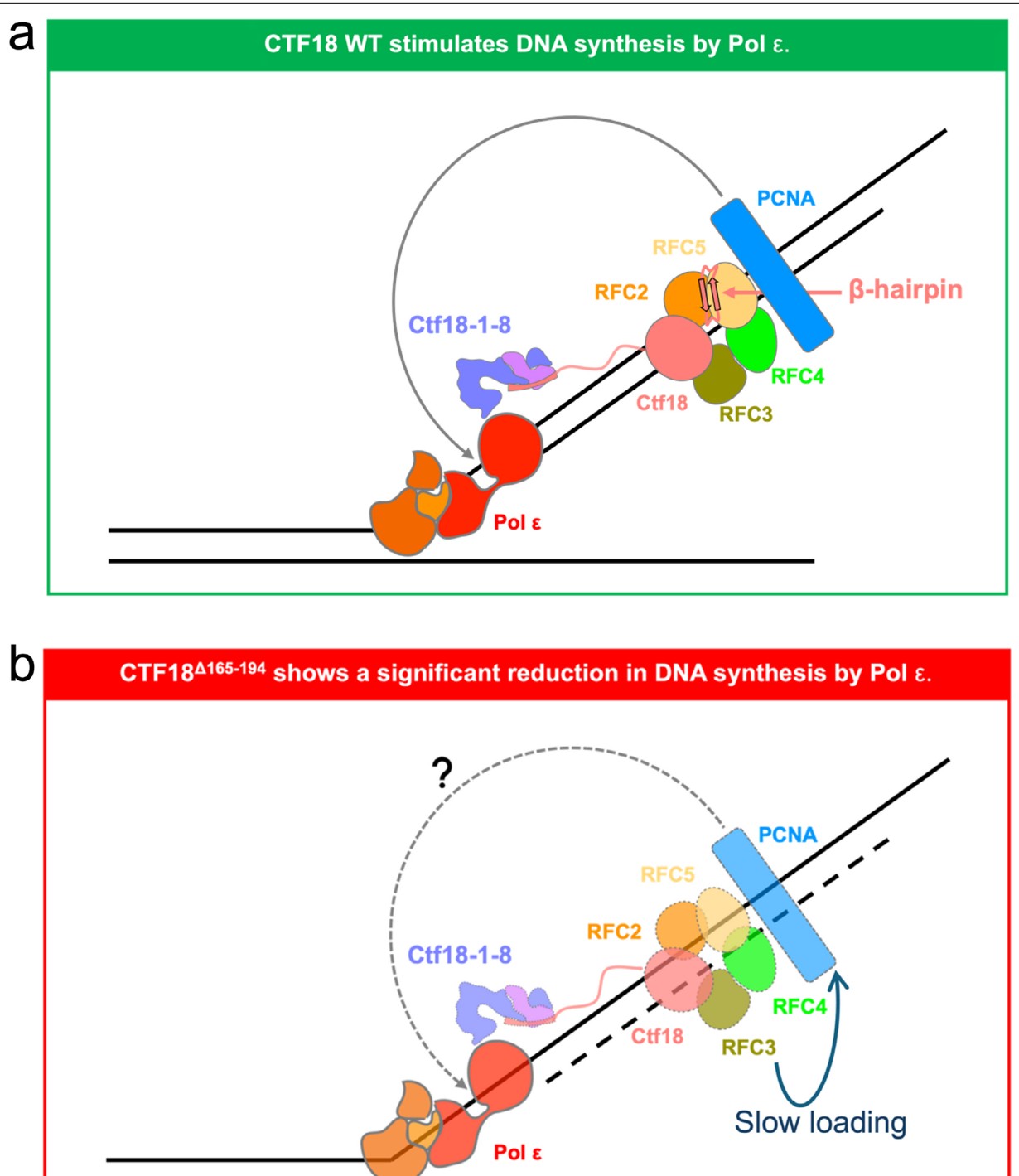

**Figure 8.** Deletion of the N-terminal β-hairpin in the Ctf18 large subunit slows down PCNA loading and reduces DNA synthesis by Pol ε. (**a**) The structure of the RFC module of CTF18–RFC is stabilized by the presence of a β-hairpin in the Ctf18 subunit (amino acids 165–194). Thus, wild-type CTF18–RFC loads PCNA onto DNA efficiently and stimulates DNA synthesis by Pol ε interacting with the polymerase. (**b**) The stability of the CTF18–RFC–PCNA complex is diminished by the deletion of the β-hairpin. Consequently, a CTF18–RFC deletion mutant lacking the β-hairpin (CTF18$^{Δ165–194}$–RFC) shows slower loading of PCNA and lower activity in stimulating primer synthesis by Pol ε.

## Materials and methods

### Protein expression and purification

Preparation of CTF18–RFC

MultiBac expression system (Geneva Biotech) was used to express human Ctf18–RFC2–5 along with two additional cohesion-specific factors, Dcc1 and Ctf8 (named hereafter CTF18–RFC) in Sf9 insect cells. For this purpose, insect cells optimized sequences of N-terminus twin Strept-tagged TEV Ctf18, RFC2, and RFC4 were cloned in pACEBac1 plasmid with independent promoters and terminators, and N-terminus 6X histidine-tagged RFC3, RFC5, Dcc1 and Ctf8 were cloned in pIDS plasmid with independent promoters and terminators by GenScript. Finally, the single transfer vector with different subunit assemblies was generated using *cre* recombinase according to the MultiBac expression system user manual. CTF18–RFC encoding all seven subunits in a single MultiBac expression plasmid (pACEBac1) was transformed into DH10MultiBac cells to make bacmid DNA. To prepare the baculovirus, bacmid DNA containing all subunits was transfected into Sf9 cells using FuGENE HD (Promega) according to the manufacturer's instructions. This baculovirus prep was amplified twice to obtain a higher titre virus (P3 virus). The expression of CTF18–RFC then proceeded by transfecting a 6 L Hi5 suspension culture at a density of $2 \times 10^6$ cells/ml with the P3 virus for 60–65 hr.

Cells were collected by centrifugation at 5500 × *g* for 10 min and re-suspended in lysis buffer [50 mM Tris-HCl (pH 7.5), 300 mM NaCl, 20 mM imidazole, 5 mM BME, 0.1% Nonidet P-40, 10% glycerol, and one EDTA-free protease inhibitor cocktail tablet/50 ml]. All further steps were performed at 4°C. Cells were sonicated and debris was removed by centrifugation at 95,834 × *g* for 1 hr at 4°C. The resultant supernatant was directly loaded onto a HisTrap HP 5 ml affinity column (Cytiva) pre-equilibrated with buffer A [50 mM Tris-HCl (pH 7.5), 300 mM NaCl, 20 mM imidazole, 5 mM BME, and 10% glycerol]. After loading, the column was washed with 50 ml of buffer A and the bound fractions were eluted by gradient with 50 ml of buffer B [50 mM Tris-HCl (pH 7.5), 300 mM NaCl, 500 mM imidazole, 5 mM BME, and 10% glycerol]. The eluents were pooled and directly loaded onto Strept Trap XT 1 ml affinity column pre-equilibrated with buffer A [50 mM Tris-HCl (pH 7.5), 300 mM NaCl, 5 mM BME, and 10% glycerol]. After loading, the column was washed with 10 ml of buffer A, followed by step elution with buffer B [50 mM Tris-HCl (pH 7.5), 300 mM NaCl, 50 mM Biotin, 5 mM BME, and 10% glycerol]. Eluents were pooled and incubated with the TEV protease overnight. Fractions were concentrated to 1 ml and loaded onto HiLoad 16/600 Superdex 200 pg pre-equilibrated with gel filtration buffer [50 mM Tris-HCl (pH 7.5), 200 mM NaCl, 1 mM DTT, and 10% glycerol]. Protein fractions were pooled in a total volume of 1.6 ml, flash frozen, and stored at –80°C.

Preparation of PCNA

To express PCNA, BL21(DE3) *E. coli* cells were transformed with WT PCNA plasmid. The transformed cells were grown at 37°C in 2YT media supplemented with ampicillin to an OD600 of 1.2. The cell cultures were induced with 0.5 mM isopropyl-beta-D-1-thiogalactopyranoside and let them grow for 19 hr at 16°C. Cells were then harvested by centrifugation at 5500 × *g* for 10 min and then re-suspended in lysis buffer [50 mM Tris-HCl (pH 7.5), 750 mM NaCl, 20 mM imidazole, 5 mM BME, 0.2% Nonidet P-40, 1 mM PMSF, 5% glycerol, and one EDTA-free protease inhibitor cocktail tablet/50 ml]. The cells were lysed with 2 mg/ml of lysozyme at 4°C for 30 min. Cells were sonicated and debris was removed by centrifugation at 22,040 × *g* for 20 min at 4°C. The cleared cell lysate was directly loaded onto a HisTrap HP 5 ml affinity column (Cytiva) pre-equilibrated with buffer A [50 mM Tris-HCl (pH 7.5), 500 mM NaCl, 20 mM imidazole, 5 mM BME, and 5% glycerol]. The column was firstly washed with 50 ml of buffer A and then with 50 ml of buffer B containing low salt [50 mM Tris-HCl (pH 7.5), 100 mM NaCl, 500 mM imidazole, 5 mM BME, and 5% glycerol]. The eluents were pooled and directly loaded onto HiTrap Q HP 5 ml anion exchange column (GE Healthcare) pre-equilibrated with buffer D [50 mM Tris-HCl (pH 7.5), 100 mM NaCl, 5 mM BME, and 5% glycerol]. The column was washed with 50 ml of buffer D after loading, followed by elution of 50 ml gradient with buffer E [50 mM Tris-HCl (pH 7.5), 1 M NaCl, 5 mM BME, and 5% glycerol]. Eluents were pooled and concentrated to 1.5 ml and loaded onto HiLoad 16/600 Superdex 200 pg pre-equilibrated with gel filtration buffer [50 mM Tris-HCl (pH 7.5), 150 mM NaCl, 1 mM DTT, and 5% glycerol]. Protein fractions were pooled, flash-frozen, and stored at –80°C. Human RPA, Cy5-PCNA, and RFC for the clamp-loading assay were obtained as previously described (*Henricksen et al., 1994*; *Hedglin et al., 2013*). The

concentration of active RPA was determined via an FRET-based activity assay as described previously (*Li et al., 2020*).

## Preparation of DNA polymerase ε

MultiBac expression system (Geneva Biotech) was used to express human DNA Pol ε in Sf9 insect cells. For this purpose, insect cells optimized the sequence of the catalytic subunit, p261, which was cloned in pACEBac1 and p59, C-terminus TEV twin Strep-tagged p17, and N-terminus 8X histidine-tagged TEV p12, along with independent promoters and terminators, were cloned in pIDS by GenScript. Finally, the single transfer vector with different subunit assemblies was generated using *cre* recombinase according to MultiBac expression system user manual. To prepare the baculovirus, the bacmid DNA of Pol ε was transfected into Sf9 cells using FuGENE HD (Promega) according to the manufacturer's instructions. The resulting supernatant was obtained as the P1 virus stock which was then amplified to obtain P2 virus stock. P2 virus stock was then further amplified to obtain P3 virus stock for large-scale expression. Pol ε was expressed by transfecting 4 L of Sf9 suspension culture at $2 \times 10^6$ cells/ml density with P3 virus. Cells were harvested after 72 hr by centrifugation at $5500 \times g$ for 10 min and re-suspended in 200 ml of lysis buffer [50 mM HEPES (pH 8), 20 mM imidazole, 400 mM NaCl, 5 mM β-Mercaptoethanol, 5% (vol/vol) glycerol, 0.1% NP-40, 1 mM PMSF, and EDTA-free protease inhibitor cocktail tablet/50 ml (Roche, UK)]. Cells were sonicated and debris was removed by centrifugation at $95,834 \times g$ for 1 hr at 4°C. The resultant supernatant was directly loaded onto a HisTrap HP 5 ml affinity column (Cytiva) equilibrated with buffer A [50 mM HEPES (pH 8), 20 mM imidazole, 400 mM NaCl, 5 mM β-Mercaptoethanol, 5% (vol/vol) glycerol, and EDTA-free protease inhibitor cocktail tablet/50 ml (Roche, UK)]. After loading, the column was washed with 50 ml of buffer A and the bound fractions were eluted by gradient with 50 ml of buffer B [50 mM HEPES (pH 8), 500 mM imidazole, 300 mM NaCl, 5 mM β-Mercaptoethanol, 5% (vol/vol) glycerol, and EDTA-free protease inhibitor cocktail tablet/50 ml (Roche, UK)]. Peak fractions containing Pol ε subunits were pooled and loaded directly onto Strep Trap XT 5 ml (Cytiva) column equilibrated with buffer C [50 mM HEPES (pH 8), 250 mM NaCl, 5 mM β-Mercaptoethanol, and 5% (vol/vol) glycerol and EDTA-free protease inhibitor cocktail tablet/50 ml (Roche, UK)]. After loading, the column was washed with 50 ml of buffer C and the bound fractions were incubated for 2–4 hr with 15 ml of 100% buffer D [50 mM HEPES (pH 8), 100 mM NaCl, 50 mM Biotin, 5 mM β-Mercaptoethanol, 5% (vol/vol) glycerol, and EDTA-free protease inhibitor cocktail tablet/50 ml (Roche, UK)] and eluted further with 30 ml of 100% buffer D. Fractions that contain all Pol ε subunits were pooled and incubated with the TEV protease for overnight and loaded onto an anion exchanger, Mono Q 5/50 GL column (Cytiva) pre-equilibrated with buffer E [50 mM HEPES (pH 8), 100 mM NaCl, 5 mM β-Mercaptoethanol, and 5% glycerol]. After loading, the column was washed with 15 ml of buffer E. The Pol ε was eluted with a 10 ml gradient from 100 mM NaCl to 1 M NaCl in 50 mM HEPES (pH 8), 5 mM β-Mercaptoethanol, and 5% glycerol. The fractions that contained all Pol ε subunits were combined and concentrated to 1 ml and loaded onto HiLoad 16/600 Superdex 200 pg (Cytiva) pre-equilibrated with gel filtration buffer [50 mM HEPES (*Bowman et al., 2004*), 200 mM NaCl, 1 mM DTT, and 5% glycerol]. Protein fractions were pooled, flash-frozen, and stored at −80°C.

All protein concentrations were calculated by measuring their absorbance at 280 nm using the extinction coefficients calculated from the amino acid composition (CTF18–RFC: 216,360 M$^{-1}$ cm$^{-1}$; PCNA: 47,790 M$^{-1}$ cm$^{-1}$, Pol ε: 351,470 M$^{-1}$ cm$^{-1}$).

## DNA substrates for cryo-EM

For the primer/template substrates used in all the structures described here, 80 base pairs template strand 5′-TTTTTTTTTTTTTTTTTTTTTTTTTTTTTTTTTTTTTTTTTTTTTTTTTTTTTTTTGACTGCAC GAATTAAGCAATTCGT AATCATGG TCATAGCT-3′ was annealed to a 39 base pairs primer containing a 3′ dideoxy cytosine chain terminator 5′-AGCTATG ACCATGATTACGAATTGCTTAATTCGTGCAGT[ddC]-3′ to form the P/T substrate. The strands were mixed in an equimolar ratio in the presence of 20 mM Tris pH 7.5 and 25 mM NaCl. The annealing reaction was performed by incubating the P/T at 92°C for 2 min followed by slow cooling to room temperature overnight. All oligonucleotides were purchased from IDT Integrated DNA Technologies.

## Clamp loading and primer extension substrates

Oligonucleotides for the clamp-loading assay comprising the 5′ddPCy3/T P/T DNA substrate were synthesized by Integrated DNA Technologies (Coralville, IA) or Bio-Synthesis (Lewisville, TX) and

purified on denaturing polyacrylamide gels. The concentrations of unlabelled DNAs were determined from the absorbance at 260 nm using the provided extinction coefficients. Concentrations of Cy3-labelled DNAs were determined from the extinction coefficient of Cy3 at 550 nm ($\varepsilon_{550}$ = 136,000 M$^{-1}$ cm$^{-1}$). For annealing the 5'ddPCy3/T P/T DNA substrate, the primer and corresponding complementary template strand were mixed in equimolar amounts in 1X annealing buffer (10 mM TrisHCl, pH 8.0, 100 mM NaCl, 1 mM EDTA), heated to 95°C for 5 min, and allowed to slowly cool to room temperature. For the prime extension assay, Oligo (592:TAACGCCAGGGTTTTCCCAGTCACG) (Integrated DNA Technologies) was annealed to 50 nM M13mp18 single-stranded DNA (New England Biolabs) in a reaction buffer consisting of [10 mM Tris-HCl (pH 7.6), 100 mM NaCl, and 5 mM EDTA]. The mixture was heated to 95°C for 5 min, then cooled gradually to room temperature. The unannealed oligonucleotide was removed using a QIAquick PCR purification kit (QIAGEN).

## Pre-steady-state FRET measurements

All experiments were performed at room temperature (23 ± 2°C) in a 16.100F-Q-10/Z15 sub-micro fluorometer cell (Starna Cells) and monitored in a Horiba Scientific Duetta-Bio fluorescence/absorbance spectrometer as described previously (*Norris et al., 2024*). In short, reaction solutions are excited at 514 nm and the fluorescence emission intensities (*I*) are monitored essentially simultaneously ($\Delta t$ = 0.118 ms) at the peak emission wavelengths for Cy3 (563 nm, $I_{563}$) and Cy5 (665 nm, $I_{665}$) over time, recording *I* every 0.17 s. For all experiments, excitation and emission slit widths are 10 nm. All recorded fluorescence emission intensities are corrected by a respective dilution factor and all-time courses are adjusted for the time between the addition of each component and the fluorescence emission intensity recordings (i.e., dead time <7 s). For any recording of the fluorescence emission intensities ($I_{665}$ and $I_{563}$), the approximate FRET efficiency is estimated from the equation $E_{FRET} = \frac{I_{665}}{I_{665}+I_{563}}$. For each experiment below, the final concentrations of all reaction components are indicated.

PCNA loading experiments were carried out in 1X Replication Buffer (25 mM HEPES, pH 7.5, 125 mM KOAc, 10 mM Mg(OAc)$_2$) supplemented with 1 mM DTT, 1 mM ATP, and the ionic strength adjusted to physiological (200 mM) by addition of KOAc. Experiments were performed via slight modifications to a published protocol (*Norris et al., 2024*; *Norris and Hedglin, 2024*). In short, 5'ddPCy3/T P/T DNA (200 nM, NeutrAvidin 800 nM homotetramer), and ATP (1 mM) are pre-incubated with RPA (600 nM heterotrimer). Then, Cy5-PCNA (200 nM homotrimer) is added, the resultant solution is transferred to a fluorometer cell, and the cell is placed in the instrument. $I_{665}$ and $I_{563}$ are monitored until both stabilize for at least 1 min. Finally, a given clamp loader complex (200 nM RFC, CTF18–RFC, or CTF18$^{\Delta165-194}$–RFC henteropentamer) is added, the resultant solution is mixed, and $I_{665}$ and $I_{563}$ are monitored, beginning <7 s after the addition of the respective clamp loader complex (i.e., 'dead time' = Dt ≤ 7 s).

## Primer extension assay

The primer extension reaction was conducted in a reaction buffer containing [40 mM HEPES (pH 7.6), 100 mM K-glutamate, 1 mM DTT, 10 mM Mg-Acetate, 200 µg/ml BSA, 1 mM ATP, 100 µM each of dATP, dTTP, and dGTP, 20 µM dCTP and 10 µCi of [$^{32}$P]-dCTP (Hartmann)]. Two nanomolar of the primed-M13 template were initially incubated at 37°C for 10 min with 100 nM PCNA, 400 nM RPA, and either CTF18–RFC WT or CTF18$^{\Delta165-194}$–RFC. The reaction was initiated by the addition of 30 nM Pol ε. Reactions were stopped at the indicated time point by adding 50 mM EDTA and electrophoresed on 0.7% alkaline agarose gel at 35 V for 17 hr. The gel was backed with DE81 paper, compressed and imaged in a Sapphire Biomolecular Imager (Azure Biosystems). All the experiments were performed in triplicate.

## Cryo-EM grid preparation and data collection

For all the complexes (WT and mutant), UltraAuFoil R1.2/1.3 Au 300 grids were glow-discharged at 30 mA for 30 s on an EasyGlow glow-discharge unit. For the preparation of the complex without Mg$^{2+}$ (**Datasets 1 and 3**), P/T DNA, ATP, CTF18–RFC WT/CTF18$^{\Delta165-194}$–RFC, and PCNA were mixed in order. The final concentrations were 5.6 mM P/T DNA, 0.2 mM ATP, 1.5 mM CTF18–RFC WT/CTF18$^{\Delta165-194}$–RFC, and 7.5 mM PCNA. The buffer this was performed in comprised 50 mM Tris-HCl pH 7.5, 200 mM NaCl, 1 mM DTT, and 0.5 mM ATP. 3 ml of the sample was applied to the grid, blotted for 2 s at blot force –5, and plunged frozen into liquid ethane using a Vitrobot Mark IV (FEI Thermo

Fisher), which was set at 4°C and 100% humidity. For the preparation of CTF18–RFC/PCNA/P-T DNA/ATP/Mg$^{2+}$ complex (**Dataset 2**), the components P/T DNA, ATP, Mg$^{2+}$, CTF18–RFC, and PCNA were mixed in order. The final concentrations were 5.6 mM P/T DNA, 0.2 mM ATP, 1.9 mg/ml Mg$^{2+}$, 1.5 mM CTF18–RFC, and 7.5 mM PCNA. The buffer this was performed in comprised 50 mM Tris-HCl pH 7.5, 200 mM NaCl, 1 mM DTT, 0.5 mM ATP, and 5 mM Mg$^{2+}$. 3 ml of the sample was applied to the grid in the same way as above for data collection. Cryo-EM data for all the samples were collected on a Thermo Fisher Scientific Titan Krios G4 transmission electron microscope at KAUST. Electron micrographs for **Dataset 1** (CTF18–RFC–PCNA with ATP in the absence of Mg$^{2+}$) were collected using a Falcon 4i detector at a dose rate of 7.5 e$^−$/pix/s for 5 s and a calibrated pixel size of 0.93 Å. Focusing was performed over a range between –2.7 and –1.5 mm, in 0.3 mm intervals. Electron micrographs for **Dataset 2** (CTF18–RFC–PCNA with ATP and Mg$^{2+}$) were collected using a Falcon 4i detector at a dose rate of 8.02 e$^−$/pix/s for 5 s and a calibrated pixel size of 0.93 Å. Focusing was performed over a range between –2.7 and –1.5 mm, in 0.3 mm intervals. Electron micrographs for **Dataset 3** (CTF18$^{\Delta165–194}$–RFC–PCNA with ATP and in the absence of Mg$^{2+}$) were collected using a Falcon 4i detector at a dose rate of 7.4 e$^−$/pix/s for 5 s and a calibrated pixel size of 0.93 Å. Focusing was performed over a range between –2.7 and –1.5 mm, in 0.3 mm intervals.

## Cryo-EM image processing

Preprocessing of all datasets (**Dataset 1**: CTF18–RFC, ATP, primer/template DNA and PCNA; **Dataset 2**: CTF18–RFC, ATP, Mg$^{2+}$, primer/template DNA and PCNA; **Dataset 3**: CTF18$^{\Delta165–194}$–RFC, ATP, primer/template DNA, and PCNA) was performed as follows: the micrographs were corrected for beam-induced motion and then integrated using MotionCor2 (*Zheng et al., 2017*). All frames were retained and a patch alignment of 4 × 4 was used. CTFFIND-4.12 estimated Contrast Transfer Function (CTF) parameters for each micrograph (*Rohou and Grigorieff, 2015*). Integrated movies were inspected with Relion-4.0 (**Datasets 1** and **2**) and Relion-5.0 (**Dataset 3**) for further image processing (7906 movies for **Dataset 1**, 6181 movies for **Dataset 2**, and 6499 movies for **Dataset 3**) (*Kimanius et al., 2021*). Particle picking was performed in automated mode using Topaz (*Bepler et al., 2019*) for **Dataset 1**. Particle extraction was carried out from micrographs using a box size of 55 pixels (pixel size: 3.65 Å/pixel). An initial dataset of 2 × 10$^6$ particles was cleaned by 2D classification followed by 3D classification with alignment. Three 3D classes were generated with populations of 31%, 28%, and 40%. 3D refinement, CtfRefine, and polishing were performed on the selected 3D classes corresponding to populations 31% and 40%. A bigger box size, 480 pixels (pixel size: 0.93 Å/pixel) was used to re-extract the particles. 3D refinement, CtfRefine, and polishing yielded reconstruction at 2.9 Å.

Particle picking was performed in reference-based mode for **Dataset 2**. Particle extraction was carried out from micrographs using a box size of 55 pixels (pixel size of 3.65 Å/pixel). An initial dataset of 3 × 10$^6$ particles was cleaned by 2D classification followed by 3D classification with alignment. Three 3D classes were generated with 30%, 33%, and 36% populations. 3D refinement, CtfRefine, and polishing were performed on the selected 3D classes corresponding to populations 33% and 36%. 3D refinement, CtfRefine, and polishing yielded reconstruction at 3.2 Å. Particle picking was performed in automated mode using Topaz (*Bepler et al., 2019*) for **Dataset 3**. Particle extraction was done from micrographs using a box size of 220 pixels (pixel size: 0.93 Å/pixel). An initial dataset of 1 × 10$^6$ particles was cleaned by 2D classification. For quantification of complex stability, the number of particles contributing to each 2D class was extracted from the classification metadata (**Datasets 1** and **3**). All classes showing isolated PCNA rings were summed and compared to the total number of particles in classes representing intact CTF18–RFC–PCNA complexes. This analysis was performed for both wild-type and β-hairpin deletion mutant datasets. Notably, no 2D classes corresponding to free PCNA were observed in the wild-type dataset, whereas in the mutant dataset, a substantial fraction of particles corresponded to isolated PCNA, suggesting reduced stability of the mutant complex.

## Cryo-EM 3D variability analysis

3DVA (*Punjani et al., 2017*; *Punjani and Fleet, 2021*) of the CTF18–RFC–PCNA consensus maps, with and without Mg$^{2+}$, was performed in CryoSPARC v 4.6 using their respective refinement masks, employing three modes and a resolution filter of 5 Å, based on global FSC resolutions of 2.9 and

3.2 Å, respectively. For each mode, three frames were generated and morphed to visualize the conformational flexibility across the components.

## Molecular modelling

The structure of the Ctf18 large subunit of human CTF18–RFC (Q8WVB6, residues 277–805) was generated with Phyre2 (*Kelley et al., 2015*), using as a template the sequence of human RFC1 (P35251, residues 581–1034). The Ctf18 subunit model was then aligned to the RFC1 subunit in the structure of the human RFC clamp loader bound to PCNA in an autoinhibited conformation (PDB: 6VVO, *Gaubitz et al., 2020*) using Chimera, the RFC1 subunit was deleted, and the resulting model including Ctf18, subunits RFC2-5 and PCNA was rigid-body fitted into the cryo-EM map. The model was edited and real-space refined with Coot (*Emsley et al., 2010*) following flexible fitting performed with ISOLDE (*Croll, 2018*). A final real-space refinement was performed in Phenix (*Adams et al., 2010*), applying secondary structure constraints. The models of both CTF18–RFC–PCNA complexes with and without the $Mg^{2+}$ ion were generated similarly.

## Acknowledgements

This research was supported by King Abdullah University of Science and Technology (KAUST) through core funding (to SMH and ADB). We thank Lingyun Zhao, Ashraf Al-Alamoudi, Alessandro Genovese, and Rachid Sougrat for their assistance at the Imaging and Characterization Core Lab at KAUST. This work was also supported by funding from the National Institutes of Health to MH (R35 GM147238-03). This publication was supported, in part, by NIH Grant T32GM149417.

## Additional information

### Funding

| Funder | Grant reference number | Author |
| --- | --- | --- |
| King Abdullah University of Science and Technology | | Giuseppina R Briola<br>Mohammad Tehseen<br>Amani Al-Amodi<br>Ammar U Danazumi<br>Phong Quoc Nguyen<br>Christos G Savva<br>Samir M Hamdan<br>Alfredo De Biasio |
| National Institutes of Health | R35 GM147238-03 | Grace Young<br>Mark Hedglin |
| National Institutes of Health | T32GM149417 | |

The funders had no role in study design, data collection, and interpretation, or the decision to submit the work for publication.

### Author contributions

Giuseppina R Briola, Mohammad Tehseen, Formal analysis, Validation, Investigation, Methodology, Writing – review and editing; Amani Al-Amodi, Formal analysis, Validation, Methodology, Writing – review and editing; Grace Young, Formal analysis, Investigation; Ammar U Danazumi, Formal analysis; Phong Quoc Nguyen, Investigation; Christos G Savva, Data curation, Methodology; Mark Hedglin, Conceptualization, Supervision, Funding acquisition, Validation, Investigation, Writing – original draft, Writing – review and editing; Samir M Hamdan, Conceptualization, Supervision, Funding acquisition, Project administration, Writing – review and editing; Alfredo De Biasio, Formal analysis, Supervision, Funding acquisition, Investigation, Methodology, Writing – original draft, Project administration, Writing – review and editing

### Author ORCIDs

Phong Quoc Nguyen https://orcid.org/0000-0001-5968-9170

Mark Hedglin [ID] https://orcid.org/0000-0003-2599-1691
Samir M Hamdan [ID] https://orcid.org/0000-0001-5192-1852
Alfredo De Biasio [ID] https://orcid.org/0000-0003-2139-2958

Reviewer #1 (Public review): https://doi.org/10.7554/eLife.103493.4.sa1
Reviewer #2 (Public review): https://doi.org/10.7554/eLife.103493.4.sa2
Reviewer #3 (Public review): https://doi.org/10.7554/eLife.103493.4.sa3
Author response https://doi.org/10.7554/eLife.103493.4.sa4

## Additional files

### Supplementary files
Supplementary file 1. Cryo-EM data collection, model refinement, and validation statistics.

MDAR checklist

### Data availability
The maps of CTF18–RFC–ATP–PCNA and CTF18–RFC–ATP–$Mg^{2+}$–PCNA complexes have been deposited in the EMBD with accession codes EMD-60534 and EMD-60598 and the atomic models in the Protein Data Bank under accession codes PDB 8ZWO and PDB 9IIN. Further atomic models used in this study are as follows: structure of human Ctf18 bound to PCNA (PDB: 6VVO, *Gaubitz et al., 2020*); structure of yeast RFC–DNA–PCNA complex (PDB: 1SXJ, *Gaubitz et al., 2022*); structure of yeast RFC–PCNA complex in open conformation (PDB: 7TKU, *Gaubitz et al., 2022*); and structure of yeast RFC–DNA–PCNA complex in open conformation (PDB: 7TIB, *Gaubitz et al., 2022*).

The following datasets were generated:

| Author(s) | Year | Dataset title | Dataset URL | Database and Identifier |
|---|---|---|---|---|
| Briola GR, Tehseen M, Al-Amodi A, Nguyen PQ, Savva CG, Hamdan SM, De Biasio A | 2025 | Structure of CTF18-PCNA with ATP | https://www.ebi.ac.uk/emdb/EMD-60534 | EMDataBank, EMD-60534 |
| Briola GR, Tehseen M, Al-Amodi A, Nguyen PQ, Savva CG, Hamdan SM, De Biasio A | 2025 | Structure of CTF18-PCNA with ATP and Mg2+ | https://www.ebi.ac.uk/emdb/EMD-60598 | EMDataBank, EMD-60598 |
| Briola GR, Tehseen M, Al-Amodi A, Nguyen PQ, Savva CG, Hamdan SM, De Biasio A | 2025 | Structure of CTF18-PCNA with ATP | https://doi.org/10.2210/pdb8zwo/pdb | Worldwide Protein Data Bank, 10.2210/pdb8zwo/pdb |
| Briola GR, Tehseen M, Al-Amodi A, Nguyen PQ, Savva CG, Hamdan SM, De Biasio A | 2025 | Structure of CTF18-PCNA with ATP and Mg2+ | https://doi.org/10.2210/pdb9iin/pdb | Worldwide Protein Data Bank, 10.2210/pdb9iin/pdb |

The following previously published datasets were used:

| Author(s) | Year | Dataset title | Dataset URL | Database and Identifier |
|---|---|---|---|---|
| Gaubitz C, Liu X, Stone NP, Kelch BA | 2024 | Structure of the human clamp loader (Replication Factor C, RFC) bound to the sliding clamp (Proliferating Cell Nuclear Antigen, PCNA) | https://doi.org/10.2210/pdb6vvo/pdb | Worldwide Protein Data Bank, 10.2210/pdb6vvo/pdb |
| Bowman GD, O'Donnell M, Kuriyan J | 2024 | Crystal Structure of the Eukaryotic Clamp Loader (Replication Factor C, RFC) Bound to the DNA Sliding Clamp (Proliferating Cell Nuclear Antigen, PCNA) | https://doi.org/10.2210/pdb1sxj/pdb | Worldwide Protein Data Bank, 10.2210/pdb1sxj/pdb |
| Gaubitz C, Liu X, Pajak J, Stone N, Hayes J, Demo G, Kelch BA | 2024 | Structure of the yeast clamp loader (Replication Factor C RFC) bound to the open sliding clamp (Proliferating Cell Nuclear Antigen PCNA) | https://doi.org/10.2210/pdb7tku/pdb | Worldwide Protein Data Bank, 10.2210/pdb7tku/pdb |
| Gaubitz C, Liu X, Pajak J, Stone N, Hayes J, Demo G, Kelch BA | 2025 | Structure of the yeast clamp loader (Replication Factor C RFC) bound to the open sliding clamp (Proliferating Cell Nuclear Antigen PCNA) and primer-template DNA | https://doi.org/10.2210/pdb7tib/pdb | Worldwide Protein Data Bank, 10.2210/pdb7tib/pdb |

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
