## [Editor Report · eLife Assessment]

This paper reports new data on the structure of the human CTF18-RFC clamp loader complex bound to the PCNA clamp. The new and **convincing** data complement previous reports of CTF-RFC-PCNA structures and as such, represents an **important** contribution.

---

## [Referee Report · Reviewer #1 (Public review)]

Summary:

The authors report the structure of the human CTF18-RFC complex bound to PCNA. Similar structures (and more) have been reported by the O'Donnell and Li labs. This study should add to our understanding of CTF18-RFC in DNA replication and clamp loaders in general. However, there are numerous major issues that I recommend the authors fix.

Strengths:

The structures reported are strong and useful for comparison with other clamp loader structures that have been reported lately.

---

## [Referee Report · Reviewer #2 (Public review)]

Summary

Briola and co-authors have performed a structural analysis of the human CTF18 clamp loader bound to PCNA. The authors purified the complexes and formed a complex in solution. They used cryo-EM to determine the structure to high resolution. The complex assumed an auto-inhibited conformation, where DNA binding is blocked, which is of regulatory importance and suggests that additional factors could be required to support PCNA loading on DNA. The authors carefully analysed the structure and compared it to RFC and related structures.

Strength & Weakness

Their overall analysis is of high quality, and they identified, among other things, a human-specific beta-hairpin in Ctf18 that flexible tethers Ctf18 to Rfc2-5. Indeed, deletion of the beta-hairpin resulted in reduced complex stability and a reduction in the rate of primer extension assay with Pol ε. Moreover, the authors identify that the Ctf18 ATP-binding domain assumes a more flexible organisation.

The data are discussed accurately and relevantly, which provides an important framework for rationalising the results.

All in all, this is a high-quality manuscript that identifies a key intermediate in CTF18-dependent clamp loading.

---

## [Referee Report · Reviewer #3 (Public review)]

Summary:

CTF18-RFC is an alternative eukaryotic PCNA sliding clamp loader which is thought to specialize in loading PCNA on the leading strand. Eukaryotic clamp loaders (RFC complexes) have an interchangeable large subunit which is responsible for their specialized functions. The authors show that the CTF18 large subunit has several features responsible for its weaker PCNA loading activity, and that the resulting weakened stability of the complex is compensated by a novel beta hairpin backside hook. The authors show this hook is required for the optimal stability and activity of the complex.

Relevance:

The structural findings are important for understanding RFC enzymology and novel ways that the widespread class of AAA ATPases can be adapted to specialized functions. A better understanding of CTF18-RFC function will also provide clarity into aspects of DNA replication, cohesion establishment and the DNA damage response.

Strengths:

The cryo-EM structures are of high quality enabling accurate modelling of the complex and providing a strong basis for analyzing differences and similarities with other RFC complexes. They use complementary pre-steady state FRET and polymerase primer extension assays to investigate the role of a unique structural element in CTF18.

Weaknesses:

The manuscript would have benefited from a more detailed biochemical analysis using mutagenesis and assays to tease apart the functional relevance of the many differences with the canonical RFC complex.

Overall appraisal:

Overall, the work presented here is solid and important. The data is sufficient to support the stated conclusions.

---

## [Author Response]

The following is the authors’ response to the previous reviews.

**Public Reviews:**

**Reviewer #1 (Public review):**
Summary:The authors report the structure of the human CTF18-RFC complex bound to PCNA. Similar structures (and more) have been reported by the O'Donnell and Li labs. This study should add to our understanding of CTF18-RFC in DNA replication and clamp loaders in general. However, there are numerous major issues that I recommend the authors fix.Strengths:The structures reported are strong and useful for comparison with other clamp loader structures that have been reported lately.Comments on revisions:The revised manuscript is greatly improved. The comparison with hRFC and the addition of direct PCNA loading data from the Hedglin group are particular highlights. I think this is a strong addition to the literature.

We thank the reviewer for their positive comments.

I only have minor comments on the revised manuscript.(1) The clamp loading kinetic data in Figure 6 would be more easily interpreted if the three graphs all had the same x axes, and if addition of RFC was t=0 rather than t=60 sec.

We now analyze and plot EFRET as a function of time after complex addition, effectively setting the loader addition to t = 0 for each trace (Figure 6 and Figs S10-14 in the new manuscript). Baseline (Ymin) and plateau (Ymax) EFRET values were obtained by averaging the stable signal regions immediately before and after clamp-loader addition, respectively. Traces are normalized to their own dynamic range before fitting.

(2) The author's statement that "CTF18-RFC displayed a slightly faster rate than RFC" seems to me a bit misleading, even though this is technically correct. The two loaders have indistinguishable rate constants for the fast phase, and RFC is a bit slower than CTF18-RFC in the slow phase. However, the data also show that RFC is overall more efficient than CTF18-RFC at loading PCNA because much more flux through the fast phase (rel amplitudes 0.73 vs 0.36). Because the slow phase represents such a reduced fraction of loading events, the slight reduction in rate constant for the slow phase doesn't impact RFC's overall loading. And because the majority of loading events are in the fast phase, RFC has a faster halftime than CTF18-RFC. (Is it known what the different phases correspond to? If it is known, it might be interesting to discuss.)

We removed the quoted statement. We avoid comparing amplitude partitions (A₁/A_T) for CTF18-RFC because (i) a substantial fraction of the reaction occurs within the <7 s dead time, and (ii) single- vs double-exponential identifiability differs across complexes. Instead, we report model-minimal progress times: RFC t_0.5_ ≤ 7 s (faster onset), CTF18-RFC ~ 8 s, CTF18^Δ165–194^-RFC ~ 12 s; completion (t_0.95_): RFC ≈ 77 s, CTF18-RFC ≈ 77 s, mutant ≈ 145 s. This shows RFC has the steeper onset, while CTF18-RFC catches up in completion, and the mutant is slower overall. We briefly note that RFC’s phases have been assigned in prior stopped-flow work and are consistent with a rapid entry step and a slower repositioning/complex release phase; we do not assign phases for CTF18-RFC here and instead rely on model-minimal timing comparisons to avoid over-interpretation.

(3) AAA+ is an acronym for "ATPases Associated with diverse cellular Activities" rather than "Adenosine Triphosphatase Associated".

Corrected to ATPases Associated with diverse cellular Activities (AAA+).

**Reviewer #2 (Public review):**
SummaryBriola and co-authors have performed a structural analysis of the human CTF18 clamp loader bound to PCNA. The authors purified the complexes and formed a complex in solution. They used cryo-EM to determine the structure to high resolution. The complex assumed an auto-inhibited conformation, where DNA binding is blocked, which is of regulatory importance and suggests that additional factors could be required to support PCNA loading on DNA. The authors carefully analysed the structure and compared it to RFC and related structures.Strength & WeaknessTheir overall analysis is of high quality, and they identified, among other things, a humanspecific beta-hairpin in Ctf18 that flexible tethers Ctf18 to Rfc2-5. Indeed, deletion of the beta-hairpin resulted in reduced complex stability and a reduction in a primer extension assay with Pol ε. Moreover, the authors identify that the Ctf18 ATP-binding domain assumes a more flexible organisation.The data are discussed accurately and relevantly, which provides an important framework for rationalising the results.All in all, this is a high-quality manuscript that identifies a key intermediate in CTF18-dependent clamp loading.Comments on revisions:The authors have done a nice job with the revision.

We thank the reviewer for their very positive comments.

**Reviewer #3 (Public review):**
Summary:CTF18-RFC is an alternative eukaryotic PCNA sliding clamp loader which is thought to specialize in loading PCNA on the leading strand. Eukaryotic clamp loaders (RFC complexes) have an interchangeable large subunit which is responsible for their specialized functions. The authors show that the CTF18 large subunit has several features responsible for its weaker PCNA loading activity, and that the resulting weakened stability of the complex is compensated by a novel beta hairpin backside hook. The authors show this hook is required for the optimal stability and activity of the complex.Relevance:The structural findings are important for understanding RFC enzymology and novel ways that the widespread class of AAA ATPases can be adapted to specialized functions. A better understanding of CTF18-RFC function will also provide clarity into aspects of DNA replication, cohesion establishment and the DNA damage response.Strengths:The cryo-EM structures are of high quality enabling accurate modelling of the complex and providing a strong basis for analyzing differences and similarities with other RFC complexes.Weaknesses:The manuscript would have benefited from a more detailed biochemical analysis using mutagenesis and assays to tease apart the differences with the canonical RFC complex. Analysis of the FRET assay could be improved.Overall appraisal:Overall, the work presented here is solid and important. The data is mostly sufficient to support the stated conclusions.

We thank the reviewer for their mainly positive assessment. Following this reviewer suggestion, we have re-analysed the FRET assay data and amended the manuscript accordingly.

Comments on revisions:While the authors addressed my previous specific concerns, they have now added a new experiment which raises new concerns.The FRET clamp loading experiments (Fig. 6) appear to be overfitted so that the fitted values are unlikely to be robust and it is difficult to know what they mean, and this is not explained in this manuscript. Specifically, the contribution of two exponentials is floated in each experiment. By eye, CTF18-RFC looks much slower than RFC1-RFC (as also shown previously in the literature) but the kinetic constants and text suggest it is faster. This is because the contribution of the fast exponential is substantially decreased, and the rate constants then compensate for this. There is a similar change in contribution of the slow and fast rates between WT CTF18 and the variant (where the data curves look the same) and this has been balanced out by a change in the rate constants, which is then interpreted as a defect. I doubt the data are strong enough to confidently fit all these co-dependent parameters, especially for CTF18, where a fast initial phase is not visible. I would recommend either removing this figure or doing a more careful and thorough analysis.

We appreciate the reviewer’s concern regarding potential overfitting of the kinetic data in Figure 6. To address this, we performed a model-minimal re-analysis designed specifically to avoid parameter covariance and over-interpretation (Figure 6 and Figs S11-14 in the new manuscript). Only data recorded after the instrument’s <7 s dead time were included in the fits, thereby excluding the partially obscured early region of the reaction. For each clamp loader complex, we selected the minimal kinetic model that produced residuals randomly distributed about zero. This approach yielded a single-exponential fit for CTF18-RFC, whereas RFC and CTF18^Δ165–194^-RFC required double-exponential fits; single-exponential models for the latter two complexes left structured residuals, clearly indicating the presence of an additional kinetic phase.

Rather than relying on co-dependent amplitude and rate parameters, we quantified the reactions by reporting progress times (t_0.5_, t_0.90_, t_0.95_), which provide a model-independent measure of reaction speed. This directly addresses the reviewer’s concern and allows a fair comparison of the relative kinetics among the complexes.

From this analysis, RFC exhibited the fastest onset (t_0.5_ ≤ 7 s; lower bound), while CTF18RFC and CTF18^Δ165–194^-RFC showed progressively slower half-times of approximately 8 s and 12 s, respectively. Completion times further emphasized these differences: both RFC and CTF18-RFC reached 95 % completion at ~77 s, whereas the mutant required ~145 s. Despite these kinetic distinctions, CTF18-RFC and its β-hairpin deletion mutant achieved similar EFRET plateaus, indicating that the mutation slows reaction progression but does not reduce the overall extent of PCNA loading.

Finally, we emphasize that our interpretation is deliberately conservative. We do not assign distinct kinetic phases to CTF18-RFC, as their molecular basis remains unresolved. RFC’s phases have been characterized in prior stopped-flow studies, but CTF18-RFC likely follows a distinct or simplified pathway. Our conclusions are thus limited to what the data unambiguously support: deletion of the Ctf18 β-hairpin decreases the rate—but not the extent—of PCNA loading, consistent with the reduced stimulation of Pol ε primer extension observed under single-turnover conditions.